# Combined Influence of Nutrient Supply Level and Tissue Mechanical Properties on Benign Tumor Growth as Revealed by Mathematical Modeling

Maxim Kuznetsov [1,2]

1   Division of Theoretical Physics, P.N. Lebedev Physical Institute of the Russian Academy of Sciences, 53 Leninskiy Prospekt, 119991 Moscow, Russia; kuznetsovmb@mail.ru
2   Nikolsky Mathematical Institute, Peoples Friendship University of Russia (RUDN University), 6 Miklukho-Maklaya Street, 117198 Moscow, Russia

**Abstract:** A continuous mathematical model of non-invasive avascular tumor growth in tissue is presented. The model considers tissue as a biphasic material, comprised of a solid matrix and interstitial fluid. The convective motion of tissue elements happens due to the gradients of stress, which change as a result of tumor cells proliferation and death. The model accounts for glucose as the crucial nutrient, supplied from the normal tissue, and can reproduce both diffusion-limited and stress-limited tumor growth. Approximate tumor growth curves are obtained semi-analytically in the limit of infinite tissue hydraulic conductivity, which implies instantaneous equalization of arising stress gradients. These growth curves correspond well to the numerical solutions and represent classical sigmoidal curves with a short initial exponential phase, subsequent almost linear growth phase and a phase with growth deceleration, in which tumor tends to reach its maximum volume. The influence of two model parameters on tumor growth curves is investigated: tissue hydraulic conductivity, which links the values of stress gradient and convective velocity of tissue phases, and tumor nutrient supply level, which corresponds to different permeability and surface area density of capillaries in the normal tissue that surrounds the tumor. In particular, it is demonstrated, that sufficiently low tissue hydraulic conductivity (intrinsic, e.g., to tumors arising from connective tissue) and sufficiently high nutrient supply can lead to formation of giant benign tumors, reaching tens of centimeters in diameter, which are indeed observed clinically.

**Keywords:** mathematical oncology; spatially distributed modeling; biomechanics; tumor mechanical properties; reaction–diffusion–convection equations; computer experiment; giant benign tumors

**MSC:** 35K57; 35Q92; 92C05

## 1. Introduction

All cancers, except for blood cancers, form solid tumors. Under favorable conditions, cancer cells should divide virtually indefinitely [1], resulting in exponential growth of tumor volume. However, exponential growth manifests itself only at the earliest stage of the tumor growth. Starting from the 1970s, experimental studies on the growth of multicellular tumor spheroids in nutrient solution have repeatedly produced S-shaped or sigmoidal growth curves [2,3]. Several factors were shown to limit the growth of spheroids. One of them is diffusional limitation of nutrient supply. It leads to the fact, that in a sufficiently large spheroid only a fraction of cells, situated in its rim, can obtain enough nutrients to proliferate. Cells, located deeper, enter a quiescent state, which requires significantly less nutrients to maintain. The radius of a spheroid, that has such structure, increases almost linearly with time. With further growth, tumor cells in the core of the spheroid begin to die under severe nutrient depletion [4]. The flow of the dead material out of the spheroid slows down its growth rate. Spheroid growth saturates when the dead material outflow—along

with other mechanisms of volume loss, like cell shedding [5]—compensates for the ongoing tumor cell proliferation.

The growth pattern of a solid tumor, growing in a tissue, is analogical in case of compact tumor growth. Such growth type is intrinsic to benign and low-stage malignant tumors. Upon tumor progression, its cells develop two main mechanisms to overcome starvation [1]. One of them is induction of angiogenesis, i.e., the formation of new capillaries in the tumor microenvironment. Angiogenesis by itself is generally considered to not change tumor growth pattern qualitatively. However, transition from avascular to vascular tumor leads to increase of its growth speed at a linear stage and to increase of its maximum volume [6]. Another mechanism is invasion of cells into surrounding tissue, which allows them to move away from nutrient-deficient regions and changes tumor growth pattern at a qualitative level. Invasion of nearby tissues is a direct indicator of malignant cancer [7].

Another growth-limiting factor, which was underestimated until the end of the 20th century, is stress-induced growth inhibition [8]. Nowadays, it is clear that the rate of tumor cells proliferation depends on the stress, exerted by the solid components of a tissue, i.e., the cells and the elements of extracellular matrix [9]. Importantly, the proliferation and death of tumor cells by themselves influence the distribution of solid stress within the tumor and surrounding normal tissue.

Solid stress can cause organ dysfunction [10], stimulate malignant tumor phenotype [11], induce resistance to chemotherapy [12] and lead to capillary compression, thus negatively affecting the delivery of drugs to the tumor [13]. Mathematical modeling of tumor growth with account of solid stress, which can, in particular, address these issues [14–16], is of significant theoretical and practical interest. In the first works in this area, tumor tissue is considered as a liquid-like medium [17,18] or an anisotropic linearly elastic medium. In the latter case, the mechanical stress tensor [19,20] is introduced into the model. Despite the relative simplicity of this approach, its use allows reproducing experimental observations and obtaining nontrivial hypotheses of a qualitative nature. For example, the work [20] reproduced the effect of reduction of the maximum size of the tumor under the increase of the externally applied pressure. The work [21] predicted the possibility of reaching the final tumor size in a non-monotonic manner, through damped oscillations.

Recently, more complex approaches have appeared, adapted from the area of solid mechanics. The works [15,22] are based on the multiplicative decomposition of tissue deformation gradient tensor into two components. They correspond to the tumor proliferation, described as the stretching of the tumor tissue, and to the elastic response of tumor and normal tissue. The stresses and deformations, occurring in them, are usually described using the neo-Hookean model for hyperelastic materials. Some works include a third component of the deformation gradient tensor. In the work [23], it corresponds to the formation of residual stress in the tumor, and the works [24,25] use it to account for reorganization of intercellular adhesive bonds in response to sufficient deformation.

Models, based on approaches from solid mechanics, can yield more realistic reproduction of solid stress distribution within the tissue [23] and can even provide quantitative predictions that are consistent with experimental results [26]. However, their solution is associated with much greater computational costs and they are not amenable to analytical investigation, like simpler approaches [20]. Importantly, both types of approaches have been used in some of the works for the consideration of the interplay of cells and interstitial fluid via modeling normal and tumor tissue as biphasic medium [20,23]. It should be noted, however, that in general, modeling tumor growth with account of biomechanical properties is not a very popular area. One of the unexplored topics, to our knowledge, is the combined influence of both crucial growth-limiting factors—nutrient availability and mechanical stress—on tumor growth.

In this work, a relatively simple mathematical model is presented, that considers growth of a non-invasive avascular tumor in biphasic tissue. The model is described in Section 2. It allows for semi-analytical investigation of tumor growth in the limit of infinite tissue hydraulic conductivity, which is performed in Section 3. The influence of combined

variation of tissue hydraulic conductivity and nutrient supply level on tumor growth is studied in Section 4. Section 5 contains discussion of the results from the biological point of view and indications of the future directions of the model development.

## 2. Model

### 2.1. Full System

The model, introduced herein, considers spherically-symmetrical growth of non-invasive avascular tumor in normal tissue. Tumor and normal tissue consist of porous solid matrix with volume fraction $c(r,t)$, and interstitial fluid, which is able to flow through the pores, with volume fraction $f(r,t)$. Here, $r$ and $t$ are independent variables of radial coordinate and time. Solid phase can be locally produced, e.g., by tumor cell proliferation with the use of interstitial fluid and substances dissolved in it as a source of mass. Destruction of solid phase, e.g., cell death, happens with its transfer into fluid phase. The rates of these processes depend on the local level of glucose $g(r,t)$ and on the local solid stress $\sigma(r,t)$. The choice of glucose as the crucial nutrient is due to its major energetic role in tumor cell metabolism and its indispensability for biosynthesis [27]. For simplicity, it is assumed that solid stress is isotropic, i.e., it acts with equal magnitude in all directions. Thus, solid phase behaves like an elastic fluid-like substance. Under assumption that the densities of two phases are equal and constant, this results in the following general equations for dynamics of tissue phases:

$$
\begin{aligned}
\text{solid phase:}\quad \frac{\partial c}{\partial t} &= \overbrace{F(c,g,\sigma)}^{\substack{\text{production/}\\\text{destruction}}} - \overbrace{\frac{1}{r^2}\frac{\partial(I_c c r^2)}{\partial r}}^{\text{convection}}, \\
\text{interstitial fluid:}\quad \frac{\partial f}{\partial t} &= \overbrace{-F(c,g,\sigma)}^{\substack{\text{production/}\\\text{destruction}}} - \overbrace{\frac{1}{r^2}\frac{\partial(I_f f r^2)}{\partial r}}^{\text{convection}},
\end{aligned}
\tag{1}
$$

where $I_f = I_f(r,t)$ denotes the velocity of convective motion of interstitial fluid through the pores of the solid phase and $I_c = I_c(r,t)$ denotes the solid phase convective velocity. Further, the tissue is considered to be fully saturated, i.e., the two phases constitute its entire volume, which is normalized to unity: $c + f = 1$. Upon summing the two equations for phases dynamics, the resulting left hand side nullifies as the derivative of a constant, and the kinetic terms cancel out due to the mass conservation. Thus, the resulting convection term is equal to zero, which yields the following relation for the velocities of tissue phases:

$$
I_c c = -I_f f.
\tag{2}
$$

Under assumptions that all external body forces (e.g., gravity) are negligible, the interstitial fluid flow through the pores is slow enough for the inertial effects to become insignificant and the solid phase is homogeneously permeable, Darcy's law can be used to obtain the rate of volumetric flow of interstitial fluid the following way:

$$
f(I_f - I_c) = -\frac{K}{\mu}\frac{\partial p}{\partial r},
\tag{3}
$$

where $K$ is the solid phase permeability, $\mu$ is the fluid viscosity, $p = p(r,t)$ is the fluid pressure. It follows from the porous media theory [28] that the gradients of fluid pressure and solid stress can be linked the following way:

$$
\frac{\partial p}{\partial r} = -\frac{\partial \sigma}{\partial r},
\tag{4}
$$

where the assumption that solid stress is isotropic is accounted for (see the work [9] where analogous approach is used). Equations (2)–(4), together with the condition of fully saturated tissue, yield

$$I_c = -\frac{K}{\mu}\frac{\partial \sigma}{\partial r}. \tag{5}$$

In general, the permeability of the solid phase $K$ should depend on its volume fraction. However, such dependence is neglected herein for two reasons. Firstly, in this work only dynamics of cells as solid matrix constituents are accounted for explicitly, while extracellular matrix dynamics are not considered. However, the implicit variation of extracellular matrix density is assumed to significantly influence the value of $K$. Secondly, as it will be shown in Section 4, in the considered system, the local fraction of tumor cells mostly varies in rather moderate range ($\approx$80–120% of its initial value), except for the case of the formation of a fluid-filled core, within which, however, the fluid does not move, and therefore the law of its movement does not need to be corrected.

Tumor cells and normal cells are assumed to have equal densities, their volumetric fractions are $n(r,t)$ and $h(r,t)$ correspondingly. Tumor cells consume glucose, and they can proliferate and die. The only component of the dynamics of normal cells is passive convective motion, analogous to that of tumor cells.

The full system of equations, that accounts for the abovementioned assumptions, is given below. Note that the equation for the interstitial fluid dynamics does not need to be considered explicitly.

$$\text{tumor cells:}\quad \frac{\partial n}{\partial t} = \overbrace{Bn \cdot \Theta_p(g) \cdot \Theta_\sigma(\sigma)}^{\text{proliferation}} \overbrace{-Mn \cdot \Theta_d(g)}^{\text{death}} \overbrace{-\frac{1}{r^2}\frac{\partial(I_c n r^2)}{\partial r}}^{\text{convection}},$$

$$\text{normal cells:}\quad \frac{\partial h}{\partial t} = \overbrace{-\frac{1}{r^2}\frac{\partial(I_c h r^2)}{\partial r}}^{\text{convection}},$$

$$\text{glucose:}\quad \frac{\partial g}{\partial t} = \overbrace{Ph[1-g]}^{\text{inflow}} + \overbrace{\frac{D_g}{r^2}\frac{\partial^2(g r^2)}{\partial r^2}}^{\text{diffusion}} \overbrace{-Q_p n \cdot \Theta_p(g) \cdot \Theta_\sigma(\sigma)}^{\text{consumption by proliferating cells}}$$

$$\overbrace{-Q_q n \cdot \{[1 - \Theta_p(g)] \cdot \Theta_\sigma(\sigma) + [1 - \Theta_\sigma(\sigma)]\} \cdot [1 - \Theta_d(g)]}^{\text{consumption by quiescent cells}},$$

$$\text{cells velocity:}\quad I_c = -\frac{K}{\mu}\frac{\partial \sigma}{\partial r},$$

$$\text{solid stress:}\quad \sigma \equiv \sigma(c) = \begin{cases} 0, & c \leq c_s, \\ k\dfrac{[c-c_0]\cdot[c-c_s]^2}{[c_0-c_s]^2}, & c_s < c < c_0, \\ k[c-c_0], & c \geq c_0, \end{cases}$$

$$\text{where}\quad \Theta_p(g) = [1 + \tanh(\epsilon\{g - g_p\})]/2,$$
$$\Theta_d(g) = [1 + \tanh(\epsilon\{g_d - g\})]/2,$$
$$\Theta_\sigma(\sigma) = [1 + \tanh(\epsilon\{\sigma_{cr} - \sigma\})]/2,$$

$$c = n + h.$$

$$(6)$$

The model accounts for two tumor growth-limiting factors discussed in Section 1, i.e., tumor cells can proliferate only under sufficient level of glucose and sufficiently low solid stress. Otherwise they become quiescent, unless glucose level is not too low. In this case, they die due to nutrient deficiency. Herein for the sake of analytical study, quite simple forms of dependencies of tumor cell proliferation and death rates on glucose level and solid stress are considered, i.e., smooth approximations of the Heaviside function. All model parameters are positive. The level of glucose needed for the tumor cells proliferation is higher than that needed for their survival, i.e., $g_p > g_d$. Since the vast majority of glucose flows into tissue by diffusion through the walls of capillaries [29], only this type of transvascular transport is considered in the model. The capillaries are assumed to rapidly

lose their functionality and degrade within the tumor due to their rupture as a result of displacement and due to chemical reasons [30]. In light of this and since angiogenesis is not considered, the surface area density of capillaries is simply taken to be proportional to the fraction of normal cells, the coefficient of proportionality being implicitly accounted for in the parameter $P$. Proliferating tumor cells consume glucose faster than quiescent cells, since they use it as a substrate for biosynthesis, i.e., $Q_p > Q_q$. Glucose consumption by normal cells is neglected for simplicity.

The function of the solid stress is based on the quite general assumption that the spatial variability at the microscale can be neglected and the volume fraction of cells can be chosen as a surrogate for the distance between them [9,20]. This function is illustrated in Figure 1. It is a smooth function, which corresponds to experimental observations at a qualitative level [31]. It assumes that at normal cell fraction, i.e., $c = c_0$, cells interaction yields zero solid stress. If cells are pushed together, $c > c_0$, the repulsive interaction appears, i.e., $\sigma > 0$. With an increase in the distance between cells, at $c < c_0$, the adhesive interaction manifests itself. At first it strengthens, and then gradually weakens due to the sequential ruptures of individual intercellular contacts. The stress eventually becomes zero at $c = c_s < c_0$. For the sake of analytical simplicity, the part of the function that corresponds to the repulsive interaction, is taken to be linear. In order for the critical fraction of cells, at which cell proliferation ceases, to be less than one, the condition $\sigma_{cr} < k[1 - c_0]$ must be met.

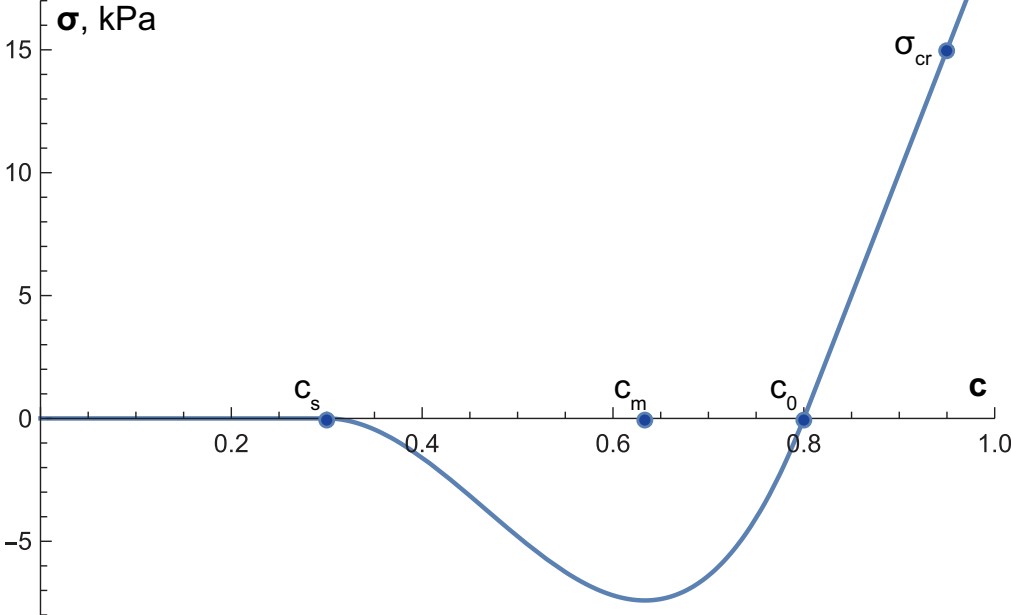

**Figure 1.** Dependence of solid stress $\sigma$ on cell fraction $c$, used in Equation (6). If cell fraction is greater than its normal value $c_0$, the repulsive interaction appears. If $c < c_0$, the adhesive interaction at first strengthens and then weakens with the decrease of cell fraction. For sufficiently sparsely distributed cells at $c < c_s$ there is no interaction between them. The parameter values are based on the basic parameter set, listed below in Table 1. $c_m \equiv [2c_0 + c_s]/3$ is the cell fraction at which solid stress is minimal.

### 2.2. Infinite Hydraulic Conductivity Limit

An important approximation, that significantly simplifies analytical tractability, is the case of infinite solid phase permeability $K$ and/or infinitesimal fluid viscosity $\mu$. Since these two parameters are used in the work only together as the fraction $K/\mu$, it will be further referred to as a single parameter of tissue hydraulic conductivity. At this limit arising infinitesimal gradients of solid stress lead to instantaneous redistribution of tissue phases and equalization of solid stress. Thus, the fractions of cells and fluid in tissue can be only equal to $c_0$ and $f_0 = 1 - c_0$. The equation of cells velocity in the system of Equation (6) degenerates, instead it can be found by summing up first two equations of this system, $\partial[n + h]/\partial t$ being taken equal to zero. This results in the following system of equations:

$$\text{tumor cells:} \quad \frac{\partial n}{\partial t} = \overbrace{Bn \cdot \Theta_p(g)}^{\text{proliferation}} - \overbrace{Mn \cdot \Theta_d(g)}^{\text{death}} - \overbrace{\frac{1}{r^2}\frac{\partial(I_c n r^2)}{\partial r}}^{\text{convection}},$$

$$\text{normal cells:} \quad \frac{\partial h}{\partial t} = -\overbrace{\frac{1}{r^2}\frac{\partial(I_c h r^2)}{\partial r}}^{\text{convection}},$$

$$\text{glucose:} \quad \frac{\partial g}{\partial t} = \overbrace{Ph[1-g]}^{\text{inflow}} + \overbrace{\frac{D_g}{r^2}\frac{\partial^2(g r^2)}{\partial r^2}}^{\text{diffusion}} - \overbrace{Q_p n \cdot \Theta_p(g) - Q_q n \cdot [1 - \Theta_p(g)] \cdot [1 - \Theta_d(g)]}^{\text{consumption by proliferating and quiescent cells}}, \quad (7)$$

$$\text{cells velocity:} \quad \frac{1}{r^2}\frac{\partial(I_c r^2)}{\partial r} = \frac{1}{c_0}[Bn \cdot \Theta_p(g) - Mn \cdot \Theta_d(g)],$$

where
$$\Theta_p(g) = [1 + \tanh(\epsilon\{g - g_p\})]/2,$$
$$\Theta_d(g) = [1 + \tanh(\epsilon\{g_d - g\})]/2.$$

Such approximation has been used in the previous works of our research group [32–35]. The conditions, under which this approximation is applicable, will be discussed in Section 4.1.

### 2.3. Parameters

The parameters of the model were estimated according to the data of various experimental works. The basic set of parameters is listed in Table 1. The dimensionless model values of parameters are the approximations of their normalized values, which were obtained with the use of the following normalization parameters: $t_n = 1$ h for time, $x_n = 10^{-2}$ cm for length, $g_n = 1$ mg/ml for glucose concentration (note that the molecular mass of glucose is 180 g/mol), $\sigma_n = 1$ kPa for stress and $n_n = 3 \cdot 10^8$ cells/ml for maximum density of tumor cells. The latter value was taken from the experimental work on growth of multicellular tumor spheroids [36]. The values for the proliferation rate of tumor cells and their glucose consumption rate were taken close to their values at the initial stage of spheroid growth according to the data of this work. Tumor cell doubling time is thus approximately a day. Tumor cells death rate was roughly taken to be an order of magnitude lower, since, firstly, experimental data suggests than across various cell lines, more than half of tumor cells survive after about two days of complete starvation [37]; secondly, under lack of glucose, tumor cells are able to prolong their survival by using other nutrients, including lactate, generated by other cells via metabolism of glucose [38].

**Table 1.** Basic set of model parameters.

| Parameter | Description | Estimated Value | Model Value | Based On |
|---|---|---|---|---|
| **For the full model and the $K/\mu \to \infty$ limit:** | | | | |
| $B$ | tumor cells proliferation rate | $0.03\ \mathrm{h}^{-1}$ | 0.03 | [36] |
| $M$ | tumor cells death rate | $0.003\ \mathrm{h}^{-1}$ | 0.003 | [37] + see text |
| $P$ | nutrient supply level | $1.1 \times 10^{-3}\ \mathrm{s}^{-1}$ | 4 & varied | [29] + see text |
| $D_g$ | glucose diffusion coefficient | $2.8 \times 10^{-6}\ \mathrm{cm}^2/\mathrm{s}$ | 100 | [39] |
| $Q_p$ | rate of glucose consumption by proliferating cells | $1.2 \times 10^{-16}\ \frac{\mathrm{mol}}{\mathrm{cells \cdot s}}$ | 24 | [36] |
| $Q_q$ | rate of glucose consumption by quiescent cells | $3 \times 10^{-18}\ \frac{\mathrm{mol}}{\mathrm{cells \cdot s}}$ | 0.5 | see text |
| $c_0$ | initial fraction of cells | 0.8 | 0.8 | [20] |
| $g_p$ | critical level of glucose for tumor cell proliferation | 0.55 mM | 0.1 | see text |
| $g_d$ | critical level of glucose for tumor cell survival | 0.055 mM | 0.01 | see text |
| $\epsilon$ | Heaviside smoothing parameter | – | 1000 (numerical)/ ∞ (analytical) | see text |
| $R_0^T$ | initial radius of tissue region | 1 cm | 100 (num. + an. full)/ ∞ (an. $K/\mu \to \infty$ limit) | see text |
| **Only for the full model:** | | | | |
| $\sigma_{cr}$ | critical stress for cell proliferation | 15 kPa | 15 | [9] + see text |
| $K/\mu$ | tissue hydraulic conductivity | $10^{-8}\ \frac{\mathrm{cm}^2}{\mathrm{mmHg \cdot s}}$ | 3 & varied | [40] + see text |
| $k$ | solid stress function coefficient | 100 kPa | 100 | [9] |
| $c_s$ | minimum cell-sensing cell fraction | 0.3 | 0.3 | [20] |

The value of $P$ has the physical meaning of microvasculature permeability surface area product. It thus was assessed as the product of two values, based on experimental data: capillary surface area density $100\ \mathrm{cm}^2/\mathrm{cm}^3$, which is close to its average value for the human muscle, and the permeability of continuous capillaries for glucose $1.1 \times 10^{-5}$ cm/s (this is the most abundant type of capillaries, present in muscles as well). Parameter $P$ was varied to study the influence of nutrient supply level on tumor growth, since it varies in different normal tissues depending on their metabolic demand. Variation of this parameter may also reflect angiogenesis and antiangiogenic therapy, since during tumor angiogenesis, both density of microvasculature and permeability of capillaries walls increase [41]. However, such reasoning assumes that capillaries do not grow into tumors, and the properties of microvasculature change equally throughout the normal tissue (see the work [34]). The highest value of $P$ was chosen to be 16 times greater than its basic value, this coefficient being chosen as a very approximate product of two estimated factors. Firstly, experiments in mouse tumor models showed that the local density of microvessels near a tumor can increase up to six times [42]. Secondly, in the work [32], it was shown that a 2.5-fold increase in the permeability of tumor capillaries to glucose should be a physiologically reasonable estimation.

The assessment of glucose consumption rate by quiescent tumor cells was based on the observation, made in the work [43], that it should be at least 40 times lower than the glucose consumption rate by proliferating cells. The values of parameters $g_p$ and $g_d$ cannot be assessed straightforwardly, since the functions of the dependence of tumor cell proliferation and death rates on glucose were chosen for phenomenological reasons. The values of these parameters were roughly assessed to differ by an order of magnitude. The use of infinite value of Heaviside smoothing parameter $\epsilon$ significantly simplifies analytical estimations, but leads to non-monotonic behavior of variable profiles during numerical simulations. Therefore, its value was adjusted for them in order to diminish this effect. Of note, such a large value of $\epsilon$ allowed keeping the concentrations of glucose positive throughout the tumor and to keep the maximum fraction of cells not approaching unity. The basic value of initial radius of normal tissue region was chosen to be significantly low to keep the cost of numerical simulations tolerable, but significantly high for the nutrient inflow to be only slightly affected during the tumor growth in comparison to infinitely

large normal tissue (except for the particular case of the final stage of growth of giant benign tumors, discussed in Sections 4.3 and 4.4).

The value of the critical stress for cell proliferation $\sigma_{cr}$ was roughly assessed by the extrapolation of the data of the work [9], in which a set of experiments was performed on growth of tumor spheroids, subjected to varying external mechanical pressure. Moreover, the maximal experimentally measured values of growth-induced solid stress in tumors, grown in mice, as well suggest that tumor cell proliferation should stop at approximately the same stress level [44]. Under the chosen value of $\sigma_{cr}$, the critical fraction of cells, at which cell proliferation ceases, is $c_{cr} = 0.95$. Importantly, experimental data suggests that in order for hydrostatic pressure alone to stop cell proliferation, it should be about three orders of magnitude higher than $\sigma_{cr}$ [45], which is the reason that its influence on cell proliferation is neglected herein. The basic value of tissue hydraulic conductivity $K/\mu$ is based on experimental measurements, performed on tumors grown in mice in the work [40]. As shown in this work, this parameter varies significantly between tumors of various cell lines and can be at least up to two orders of magnitude higher. However, other experimental data suggest that it can also be at least two orders of magnitude lower, e.g., in case of fibrosarcoma [46]. This is a type of tumor, derived from fibrous connective tissue and characterized by the abundance of proliferating fibroblasts. These cells produce the elements of extracellular matrix, thus decreasing the permeability of the solid phase of the tissue. The influence of variation of tissue hydraulic conductivity on tumor growth will be discussed in Section 4.

*2.4. Numerical Solving*

During numerical simulations, the sets of Equations (6) and (7) were solved in a region with initial size $R_0^T$, which further increased due to the tumor growth. The following initial conditions were used, which represent a normal tissue with a small spherical colony of tumor cells with radius $R_0 = 0.1$ mm, located in its center, where $r = 0$:

$$\begin{cases} n = c_0, \\ h = 0, \quad for\ r \leq R_0; \\ g = 1 \end{cases} \qquad \begin{cases} n = 0, \\ h = c_0, \quad for\ r > R_0. \\ g = 1 \end{cases} \tag{8}$$

For cells, a zero-flux boundary condition was used at the center, and constant values $n = 0$ and $h = c_0$ were used at the moving tissue boundary. Of note, this condition implied the constancy of solid stress and interstitial fluid pressure there. Furthermore, it implied the inflow of fluid through the moving normal tissue boundary that eventually provided the overall increase of mass during the tumor growth. For glucose, a zero-flux boundary condition was used at both edges. In case of infinite hydraulic conductivity limit, the convective velocity was set to zero at the tumor center, resulting in the following equation for it:

$$I_c(r,t) = \frac{1}{r^2} \int_0^r \frac{z^2}{c_0} [Bn(z,t) \cdot \Theta_p(g(z,t)) - Mn(z,t) \cdot \Theta_d(g(z,t))] \mathrm{d}z. \tag{9}$$

The method of splitting into physical processes was used for all variables, i.e., kinetic equations, diffusion equation and convective equations were solved successively during each time step. Implicit Crank–Nicholson scheme was used for glucose diffusion equation. Convective equations were solved using the conservative flux-corrected transport algorithm with implicit anti-diffusion stage. Importantly, this method by itself nevertheless introduces a small amount of irremovable diffusion, which leads to artificial invasion of tumor into normal tissue. Another problem of straightforward implementation of this method is the impossibility to correctly simulate the movement of the free tissue boundary on a uniform space grid. In order to overcome these problems, two additional floating grid points were introduced to indicate the interface between the tumor and normal tissue and the location of the free tissue boundary. The positions of these grid points were calculated

on the basis of cell volume conservation during the solution of convective equations at each time step.

During the solution of the full model, variation of the hydraulic conductivity parameter affected the stability of the numerical algorithm, its greater values demanding a lower time step. Therefore, time step was dynamically adjusted by empirical methods in order to keep solutions stable. Its maximum value was restricted by a glucose diffusion coefficient, which also defined its choice during the solution of the model in the infinite hydraulic conductivity limit. The correctness of choice of time and space steps followed from the comparison of numerical and analytical solutions, presented in Sections 3.4 and 4.1. Since time step was defined by convective equations and sometimes glucose diffusion coefficient, it was decided to solve kinetic equations by a relatively simple explicit Euler method.

Flux-corrected transport algorithm was introduced in the work [47], while other classical methods are described in many books (see, e.g., [48]). The computational code was implemented in C++ and can be found in the Supplementary Materials.

## 3. Estimation of Growth Curves in the Limit of Infinite Tissue Hydraulic Conductivity

The system of Equation (7) allows for semi-analytically derived approximations of the tumor growth curves, produced by the full system of Equation (6) in case of sufficiently large values of $K/\mu$. As it was noted in Section 2.2, in this case the fraction of both tumor and normal cells in tissue can only be equal to $c_0$. Since a non-invasive tumor was considered, there always exists a clear border between tumor and normal tissue. Its location in space will be denoted as $R = R(t)$. Moreover, in case of $\epsilon \to \infty$ from the construction of the model, it also follows that the point in space $R_p(t)$, where glucose level is $g_p$, indicates a clear border between proliferating and quiescent cells, and the point $R_d(t)$, where glucose level is $g_d$, indicates a clear border between quiescent and dying cells. Thus, in every moment of time, the solution can be split into several regions, in each of which some terms of the system of Equation (7) can be omitted or simplified. Importantly, since the tumor growth speed should be significantly lower than the characteristic speed of glucose diffusion within the tissue, quasistationary approximation can be used for estimation of the profile of glucose in every moment (i.e., setting $\partial g/\partial t = 0$). Glucose profile can be found by procedure, exemplified in the previous work [34], i.e., the so-called method of stitching of functions, that are the common analytical solutions of the reduced forms of the system of Equation (7) in every region. That is, the coefficients in these common solutions should be picked in such a way that glucose distribution turns into a piecewise function that is continuously differentiable in order to provide the continuity of glucose concentration and of its flow. Solution of such tasks allows for the assessment of tumor growth curve $R(t)$, which will be discussed in this section. The calculations, performed as a result of such an approach, are described in Appendix A. Initial conditions for the performed calculations matched Equation (8). On the left border, a zero-flux boundary condition was used for glucose. The initial size of the normal tissue $R_0^T$ was taken in this section to be infinite for simplicity, therefore, the right boundary condition for glucose was $\lim_{r \to \infty} g(r) = 1$.

### 3.1. First Growth Stage: Exponential Phase

An important moment for the tumor growth curve estimation is the moment $T_p$ : $g(0, T_p) = g_p$, at which quiescent tumor cells have just begun to appear. The distributions of the variables for this moment are shown in Figure 2. Here and further on, the basic set of parameters, listed in Table 1, is used for illustrations, unless otherwise stated.

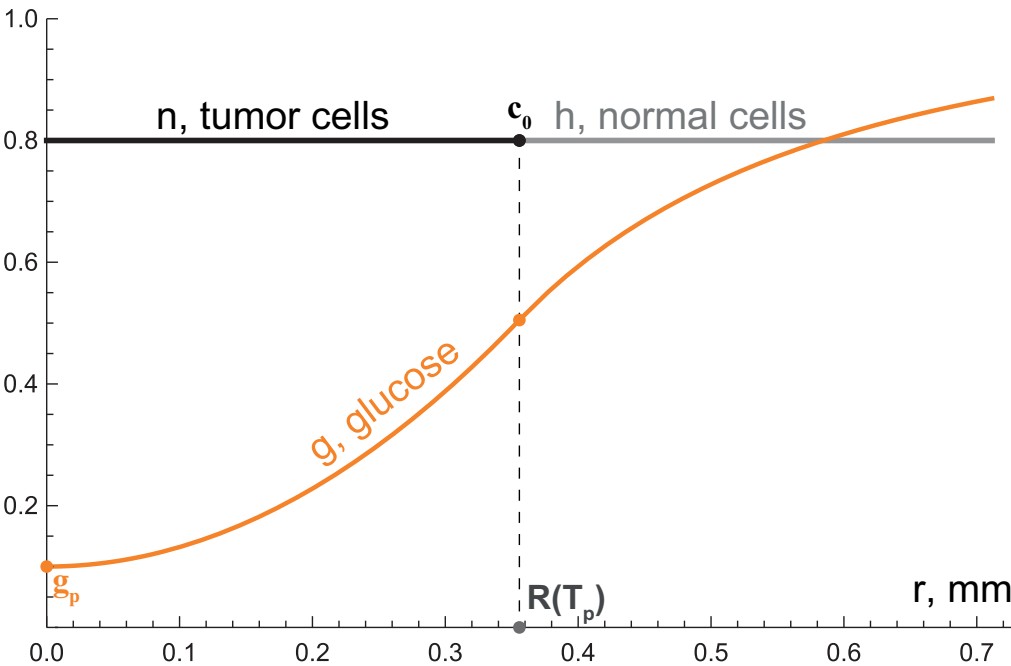

**Figure 2.** The distributions of the variables of the tumor growth model, governed by the system of Equation (6), at the moment when first quiescent tumor cells have just begun to appear. Prior to this moment, all tumor cells were constantly proliferating.

The tumor radius can be found as the only solution, yielded by stitching of two functions of glucose distributions in regions of tumor and normal tissue, that has physical meaning:

$$R(T_p) = \sqrt{\frac{D_g}{c_0 P}} \left\{ -1 + 2\sqrt{\frac{2[1-g_p]P + Q_p}{Q_p}} \right. $$
$$\left. \times \cos\left(\frac{1}{3}\left[\pi - arctg\sqrt{-1 + \frac{\{2[1-g_p]P + Q_p\}^3}{Q_p^3}}\right]\right) \right\}. \tag{10}$$

For the basic parameter set $R(T_p) \approx 0.36$ mm. Interestingly, this function has limits for both infinitesimal and infinitely large levels of nutrient supply from the normal tissue:

$$\lim_{P \to 0} R(T_p) = \sqrt{\frac{2[1-g_p]D_g}{c_0 Q_p}}, \quad \lim_{P \to \infty} R(T_p) = \sqrt{\frac{6[1-g_p]D_g}{c_0 Q_p}}, \tag{11}$$

which are $\approx$0.31 mm and $\approx$0.53 mm for the basic parameter set. Thus, for infinitely large normal tissue the maximum radius of a tumor, which consists of only proliferating cells, cannot fall below a certain threshold under an arbitrarily small level of nutrient supply from surrounding normal tissue. This is in contrast with the corresponding result for planar geometry—according to the results of the work [34], $\lim_{P \to 0} R(T_p) = 0$ for the planar case. For finite normal tissue, this limit as well equals zero in the three-dimensional case; however, a noticeable decrease of $R(T_p)$ requires a very significant decrease in nutrient supply level. For instance, if $R_0^T = 1$ cm, $P$ should be decreased about 430 times from its basic value for $R(T_p)$ to become lower than 0.3 mm.

The speed of tumor front propagation $V(T_p) \equiv \dot{R}(T_p)$, where upper dot denotes differentiation with respect to time, can be assessed via Equation (9), which, applied to $r(t) = R(t)$, corresponds to the volume or mass conservation law for the tumor cells. That is, the increase of the tumor cells volume during the infinitesimal period of time $dt$, close to $T_p$, is provided by the newborn cells, all tumor cells being in proliferative state at this period of time:

$$4\pi R(T_p)^2 \cdot c_0 V(T_p)dt = \frac{4}{3}\pi R(T_p)^3 \cdot c_0 B dt. \tag{12}$$

Therefore, $V(T_p) = BR(T_p)/3$, which is $\approx 0.60$ mm/week for the basic parameter set.

From the beginning of tumor growth up to this moment, all tumor cells proliferate, since glucose level at each point in space falls monotonically with time and $g(r,t) > g_p \ \forall r < R(t) \ \forall t < T_p$. Therefore, in the beginning of the tumor growth, its radius increases exponentially:

$$R(t) = R_0 e^{Bt/3}, \ t \le T_p = \frac{3}{B}\ln(R(T_p)/R_0), \tag{13}$$

$T_p$ being $\approx 5.3$ days for $R_0 = 0.1$ mm and the basic parameter set.

### 3.2. Second Growth Stage: Two-Layered Tumor

Figure 3 illustrates the distributions of the model variables at the moment $T_d$ : $g(0,T_d) = g_d$, when cells at the tumor center have just begun to die.

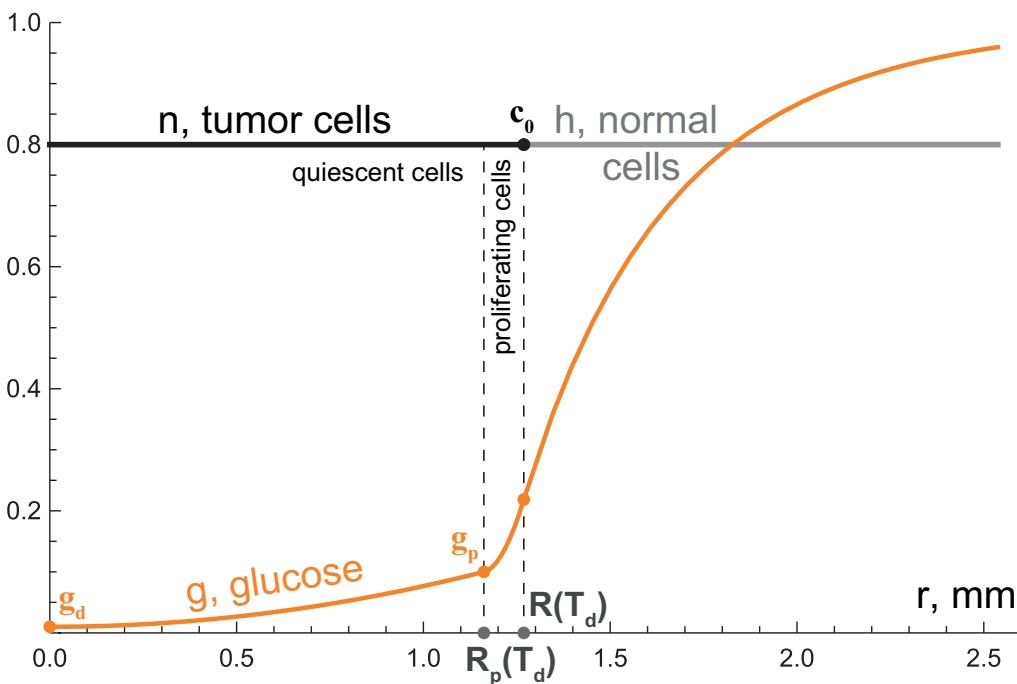

**Figure 3.** The distributions of the variables of the tumor growth model, governed by the system of Equation (6) at the moment when first tumor cells have just begun to die.

The tumor radius at this moment is:

$$
\begin{aligned}
R(T_d) = & \sqrt{\frac{D_g}{c_0 P}} \Bigg\{ -1 + 2\sqrt{\frac{2[1-g_d]PQ_q + 6[g_p - g_d]P[Q_p - Q_q] + Q_p Q_q}{Q_p Q_q}} \\
& \times \cos\left(\frac{1}{3}\left[\pi - arctg\sqrt{-1 + \frac{\{2[1-g_d]PQ_q + 6[g_p - g_d]P[Q_p - Q_q] + Q_p Q_q\}^3}{\{6\sqrt{6}[g_p - g_d]^{3/2}[PQ_p]^{3/2}[Q_p - Q_q] + [Q_p Q_q]^{3/2}\}^2}}\right]\right) \Bigg\}.
\end{aligned}
\tag{14}
$$

For the basic parameter set $R(T_d) \approx 1.27$ mm. This function also has limits for both infinitesimal and infinitely large levels of nutrient supply:

$$\lim_{P \to 0} R(T_d) = \sqrt{\frac{2D_g}{c_0 Q_p Q_q} \left\{ [1 - g_d]Q_q + 3[g_p - g_d][Q_p - Q_q] \right\}},$$

$$\lim_{P \to \infty} R(T_d) = 2\sqrt{\frac{2D_g}{c_0 Q_p Q_q} \left\{ [1 - g_d]Q_q + 3[g_p - g_d][Q_p - Q_q] \right\}}$$

$$\times \cos\left( \frac{1}{3}\left[ \pi - arctg\sqrt{-1 + \frac{\{2[1 - g_d]Q_q + 6[g_p - g_d][Q_p - Q_q]\}^3}{Q_p\{6\sqrt{6}[g_p - g_d]^{3/2}[Q_p - Q_q]\}^2}} \right] \right), \tag{15}$$

which are $\approx 1.19$ mm and $\approx 1.49$ mm for the basic parameter set.

The radius of the quiescent zone is

$$R_p(T_d) = \sqrt{\frac{6[g_p - g_d]D_g}{c_0 Q_q}}, \tag{16}$$

which, interestingly, does not depend on the nutrient supply level. For the basic parameter set $R_p(T_d) \approx 1.16$ mm. Note that the minimal width of proliferating rim thus is

$$\lim_{P \to 0}(R(T_d) - R_p(T_d)) = \sqrt{\frac{6D_g[g_p - g_d]}{c_0 Q_q} + \frac{2D_g\{[1 - g_d] - 3[g_p - g_d]\}}{c_0 Q_p}} - \sqrt{\frac{6D_g[g_p - g_d]}{c_0 Q_q}}, \tag{17}$$

which is $\approx 0.03$ mm for the basic set of parameters. If $g_p > [1 + 2g_d]/3$, this value is negative. In this case under sufficiently low nutrient supply level tumor should achieve a state in which all its cells are quiescent and stop growing before glucose level in the tumor center falls down to $g_d$.

As in Section 3.1, the speed of the tumor front propagation $V(T_d) \equiv \dot{R}(T_d)$ can be assessed via the mass conservation law:

$$V(T_d) = B\frac{R(T_d)^3 - R_p(T_d)^3}{3R(T_d)^2}, \tag{18}$$

which is $\approx 0.49$ mm/week for the basic parameter set.

Unlike the case of the exponential phase, no explicit expression for $R(t)$ can be derived for the second stage of tumor growth, at which quiescent and proliferating zones are present. This part of the growth curve, however, can be estimated numerically by using the fact that replacing $g_d$ in right hand sides of Equations (14) and (16) with $g_0 \in [g_d, g_p]$ yields the values of $R(t)$ and $R_p(t)$ for some moment of time $T_p < t < T_d$. In particular, implicit function $g_0(R)$ can be approximated from Equation (14) for $R(T_p) < R < R(T_d)$. Then, Equation (16) and mass conservation law lead to the ODE

$$\dot{R}(t) = B\frac{R(t)^3 - \{6[g_p - g_0(R(t))]D_g/(c_0 Q_q)\}^{3/2}}{3R(t)^2} \quad \forall t : T_p < t < T_d, \tag{19}$$

which can be solved numerically. The value of $T_d$ followed from its solution, for the basic set of parameters, it is $\approx 16.6$ days.

### 3.3. Third Growth Stage: Tending to Plateau

As it follows from Equation (9), the death of tumor cells contributes to the deceleration of tumor front propagation, and the larger the total volume of dying cells, the greater this contribution. At the last stage, the tumor growth tends to a complete stop, and the distributions of model variables tend to stable profiles, depicted in Figure 4, where $\hat{R} \equiv lim_{t \to \infty}R(t)$, $\hat{R}_p \equiv lim_{t \to \infty}R_p(t)$, $\hat{R}_d \equiv lim_{t \to \infty}R_d(t)$. Note that the glucose level is uniform within the core of dying cells.

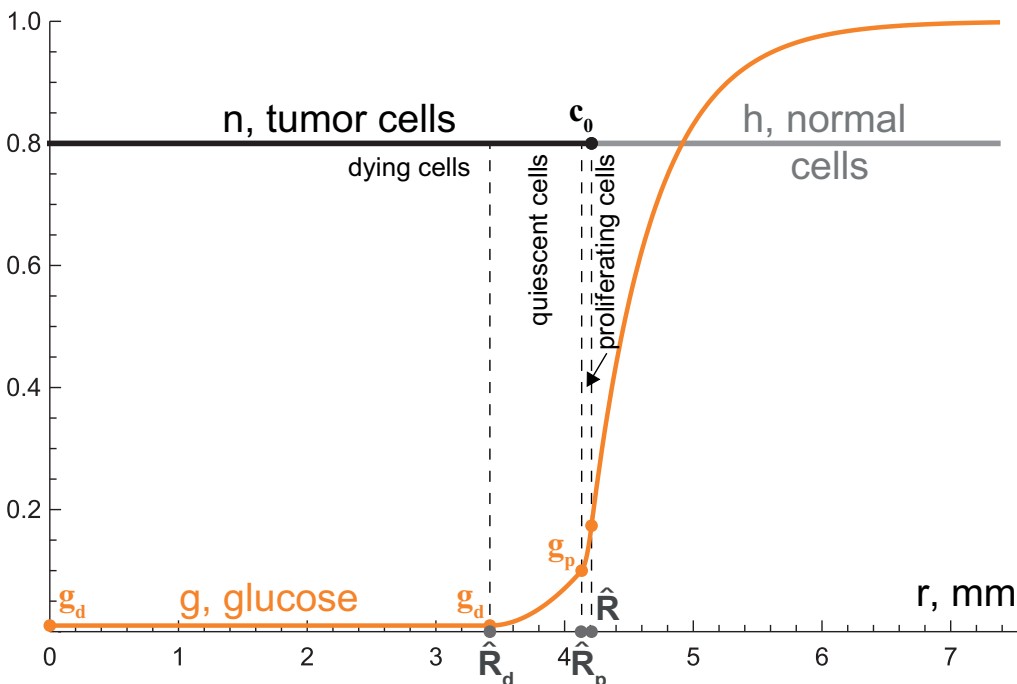

**Figure 4.** The limit stable distributions of the variables of the tumor growth model, governed by the system of Equation (6).

The values of $\hat{R}$, $\hat{R}_p$ and $\hat{R}_d$ can be, in the general case, found only numerically. The stitching of glucose distributions functions yields the following relations between them:

$$\hat{R}_p(\hat{R}_d) = \frac{f(\hat{R}_d) + g(\hat{R}_d)^{2/3}}{\sqrt{c_0 Q_q} \cdot g(\hat{R}_d)^{1/3}},$$ (20)

$$f(\hat{R}_d) = 2[g_p - g_d]D_g + [\sqrt{c_0 Q_q}\hat{R}_d]^2, \quad g(\hat{R}_d) = -[\sqrt{c_0 Q_q}\hat{R}_d]^3 + \sqrt{[\sqrt{c_0 Q_q}\hat{R}_d]^6 - f(\hat{R}_d)^3},$$

$$\hat{R}(\hat{R}_d) = \sqrt{\frac{D_g}{c_0 P}}\left\{-1 + 2\sqrt{\frac{h(\hat{R}_d)}{3D_g Q_p \hat{R}_p(\hat{R}_d)}}cos\left(\frac{1}{3}\left[\pi - arctg\sqrt{-1 + \frac{h(\hat{R}_d)^3}{l(\hat{R}_d)}}\right]\right)\right\},$$

$$h(\hat{R}_d) = 3D_g\{2[1 - g_p]P + Q_p\}\hat{R}_p(\hat{R}_d) + c_0 P\{3Q_p \hat{R}_p(\hat{R}_d)^3 - 2Q_q[\hat{R}_p(\hat{R}_d)^3 - \hat{R}_d^3]\},$$

$$l(\hat{R}_d) = 27Q_p\hat{R}_p(\hat{R}_d)^3\left[D_g^{3/2}Q_p + [c_0 P]^{3/2}\{Q_p\hat{R}_p(\hat{R}_d)^3 - Q_q[\hat{R}_p(\hat{R}_d)^3 - \hat{R}_d^3]\}\right]^2.$$ (21)

Another relation between these values is yielded by the fact that tumor front propagation speed for the stable state equals to zero. Hence, due to the mass conservation:

$$B[\hat{R}^3 - \hat{R}_p^3] = M\hat{R}_d^3.$$ (22)

The resulting system of Equations (20)–(22) can be solved numerically. For the basic set of parameters, $\hat{R}_d \approx 3.42$ mm, $\hat{R}_p \approx 4.13$ mm and $\hat{R} \approx 4.21$ mm. Furthermore, numerical simulations suggested that, under infinite nutrient supply from normal tissue, there exist limit values $lim_{P\to\infty}\hat{R}_d \approx 10.36$ mm, $lim_{P\to\infty}\hat{R}_p \approx 11.05$ mm, $lim_{P\to\infty}\hat{R} \approx 11.34$ mm. For $P = 64$, that, as it was noted in Section 2.3, was chosen as a physiologically justified limit of variation, maximum tumor radius is $\hat{R} \approx 7.87$ mm, which is $\approx 70\%$ of its limit value. There also exist the limits for infinitesimal nutrient supply level: $lim_{P\to 0}\hat{R}_d \approx 1.15$ mm, $lim_{P\to 0}\hat{R}_p \approx 1.94$ mm, $lim_{P\to 0}\hat{R} \approx 1.95$ mm.

The part of the growth curve for the last stage of tumor growth, at which dying cells are present, can be estimated numerically by using the fact that replacing $\hat{R}_d$ in the right hand sides of Equations (20) and (21) with $R_d \in [0, \hat{R}_d]$ yields the values of $R(R_d(t))$ and $R_p(R_d(t))$ for some moment of time $t > T_d$. With their use, the ODE for the radius of spheroid of dying cells can be obtained via mass conservation law:

$$\dot{R}_d(t) = \frac{B[R(R_d(t))^3 - R_p(R_d(t))^3] - MR_d(t)^3}{3R(R_d(t))^2 \cdot [\partial R(R_d)/\partial R_d]_t},$$ (23)

and inserting the solution into the right hand side of Equation (21) yields the function $R(R_d(t))$, that approximates the growth curve for $t > T_d$. For the basic set of parameters, tumor radius reaches 90% of its maximum value at $\approx 96.6$ days, and 99% of it at $\approx 191.3$ days.

### 3.4. Comparison with Numerical Simulations

Figure 5 compares the tumor growth curves for the system of Equation (7), obtained by the methodology, described in Sections 3.1–3.3, with the results of the numerical simulations of the same system, performed as described in Section 2.4. Five cases with different values of $P$ were considered, other parameters being from their basic set. As it is seen from the figure, the results, produced by two methods, are in good agreement. In every case, the discrepancy between the estimations of tumor radius does not exceed 0.5%, reaching its maximum at the beginning of the second stage of tumor growth.

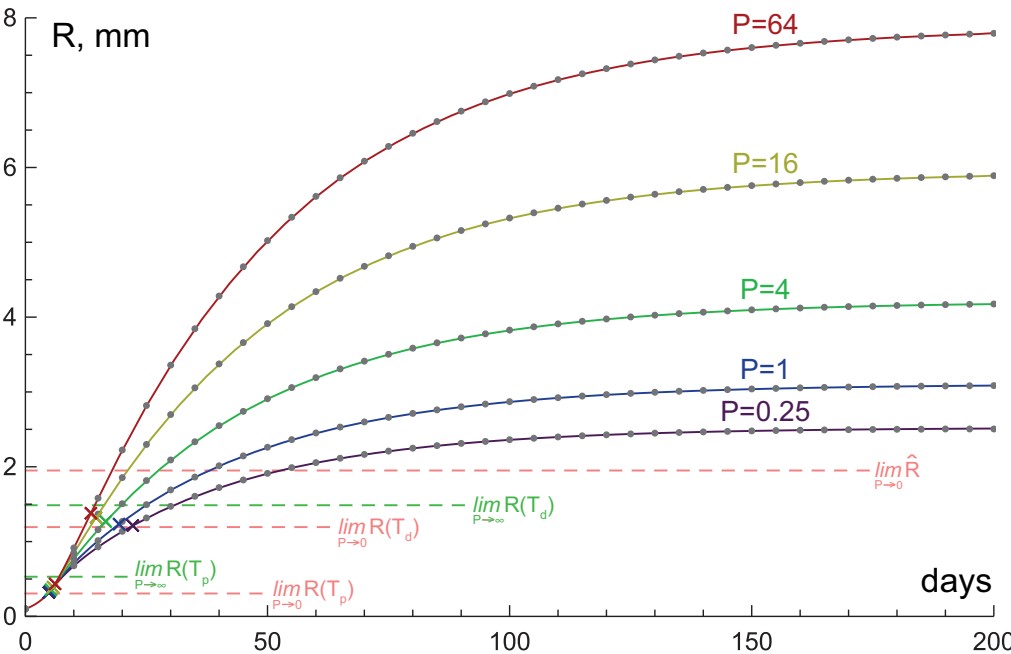

**Figure 5.** Tumor growth curves, produced by system of Equation (7) for different nutrient supply levels $P$, obtained by semi-analytical methodology (solid lines) and by numerical simulations (gray dots). Multicolored crosses denote the moments when glucose concentration in tumor center equals $g_p$, below which tumor cells turn to quiescent state, and $g_d$, below which tumor cells die. The denoted limit values for tumor radius under infinitesimal and infinitely large values of $P$ are defined by Equations (11) and (15), except for $lim_{P \to 0}\hat{R}$, which is estimated numerically, as well as $lim_{P \to \infty}\hat{R} \approx 11.34$ mm, which is not shown.

## 4. Study of Influence of Tissue Hydraulic Conductivity and Nutrient Supply Level on Tumor Growth

The dynamics of the full model, expressed by the system of Equation (6), was studied via numerical simulations, performed as described in Section 2.4, under variation of tissue hydraulic conductivity $K/\mu$ and nutrient supply level $P$. The same set of five values of $P$, as in the previous section, was used. The results are arranged in this section by decreasing values of $K/\mu$.

### 4.1. High Hydraulic Conductivity: When the Infinite Hydraulic Conductivity Limit Is Applicable

Numerical simulations with the basic value of $K/\mu = 3$ produced tumor growth patterns, visually almost indiscernible from those shown in Figure 5. Namely, simulations

of the full model produced the values of tumor radii, which up to the beginning of the third stage of tumor growth were lower by no more than 0.3%, than in the approximate model that assumed infinite hydraulic conductivity. On the third stage of growth, this difference began to decrease. Under high values of $P$, the simulations of the full model eventually resulted in slightly greater radii of tumors, than in the approximate model—by $\approx$0.3% under $P = 16$ and by $\approx$0.5% under $P = 64$. The smallness of this difference was due to the fact that such high value of hydraulic conductivity led to sufficiently fast smoothing of the solid stress gradients, caused by spatial variations of cell fractions in tissue. Indeed, even under the highest value of $P = 64$, the fractions of cells did not deviate from their initial level $c_0$ by more than $\approx$0.4% up and $\approx$1.3% down. Therefore, the results suggested that the infinite tissue hydraulic conductivity limit should provide good approximation for the full model under values of $K/\mu$ at least as great as $10^{-8}$ cm$^2$/(mmHg $\cdot$ s) within the physiologically reasonable range of nutrient supply level. Of note, each simulation of the full model with $K/\mu = 3$ lasted about 200 times longer than the corresponding simulation of the approximate model. However, under high $K/\mu$, it is the cell movement that was the most crucial factor defining the appropriate size of the time step, and with the decrease of $K/\mu$ the numerical cost noticeably decreased.

### 4.2. Intermediate Hydraulic Conductivity: Fluid-Filled Core Forms under High Nutrient Inflow Level

Figure 6 shows tumor growth curves, obtained numerically in case of $K/\mu = 0.3$. The growth curves for the already discussed case of $K/\mu = 3$ are also present for comparison as dashed lines. As it is seen from this figure, with the decrease in $K/\mu$, the change of tumor growth curves followed the same tendency, which was discussed in the previous section. Namely, tumor radii became slightly lower during the first several dozens of days of tumor growth, but eventually they became greater. The final increase in the tumor sizes was more pronounced with the increase in $P$. For the four lowest values of $P$ the decrease of $K/\mu$ by an order of magnitude led to the increase of maximum tumor radius by $\approx$0.3%, $\approx$1.0%, $\approx$1.7% and $\approx$3.5%. For the largest value of $P = 64$, it increased by $\approx$25.3%.

Figure 7 provides two snapshots of distributions of the model variables under $K/\mu = 0.3$ and $P = 16$, which allow explaining qualitatively such changes in dynamics for the lowest four values of $P$. Figure 7a corresponds to the 14th day of tumor growth, which was the last day when glucose concentration in the tumor center was greater than $g_d$, and thus tumor cells had not died yet. Under this value of $K/\mu$, the spatial variations in cell fractions smoothed out on a time scale comparable to that of tumor growth. Therefore, the fraction of cells within the tumor remained noticeably increased due to their proliferation in comparison to their initial value $c_0$. Since the tumor cells were more densely packed than in case of $K/\mu = 3$, while their overall proliferation rates were comparable, tumor had a slightly smaller size during the first days of growth in the case with the lower value of $K/\mu$.

Figure 7b corresponds to the 350-th day of tumor growth, after which further differences in profiles of variables remained indiscernible. Therefore, this snapshot may be referred to as their stable state. As the movement of the normal tissue practically ceased at this moment, the distribution of fraction of its cells was almost uniform, close to $c_0$. Due to the death of tumor cells, their fraction was close to $c_0$ only at the interface with the normal tissue and decreased towards the tumor center. Dying tumor cells were less densely packed in this case in comparison with the case of greater value of $K/\mu = 3$. Therefore, in order to stop tumor growth, significantly larger region of dying tumor cells was needed to compensate for the proliferation of cells in the tumor rims, the amounts of which were quite comparable in both cases. That resulted in greater total size of the stable tumor in the case of lower $K/\mu$.

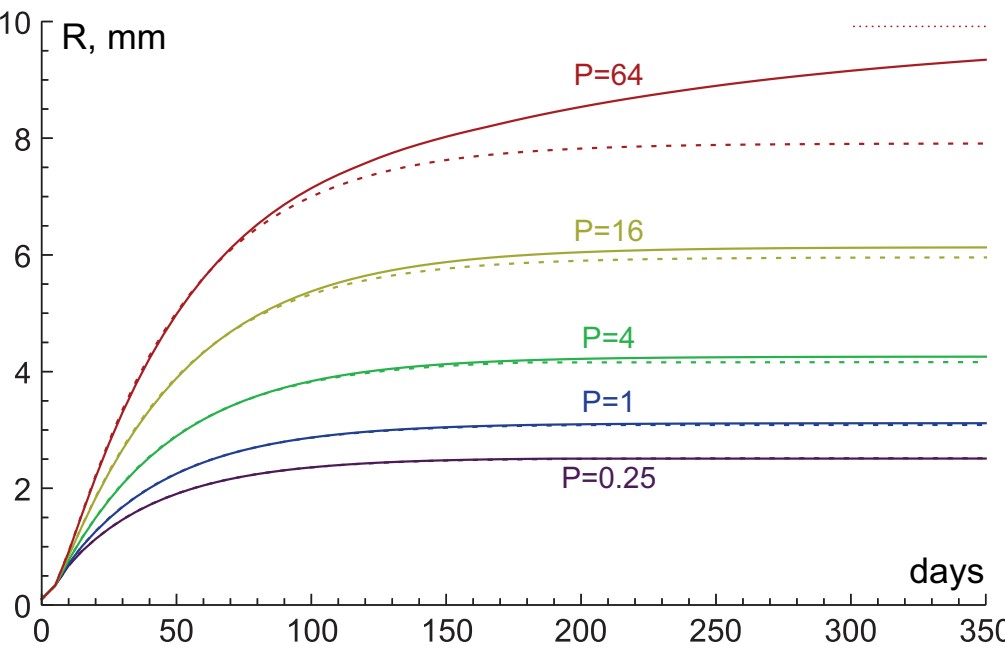

**Figure 6.** Tumor growth curves, produced by numerical simulations of the system of Equation (6) under different nutrient supply levels $P$ and tissue hydraulic conductivity $K/\mu = 0.3$ (solid lines) and $K/\mu = 3$ (dashed lines). Red dotted line denotes the limit value of tumor radius under $K/\mu = 0.3$ and $P = 64$.

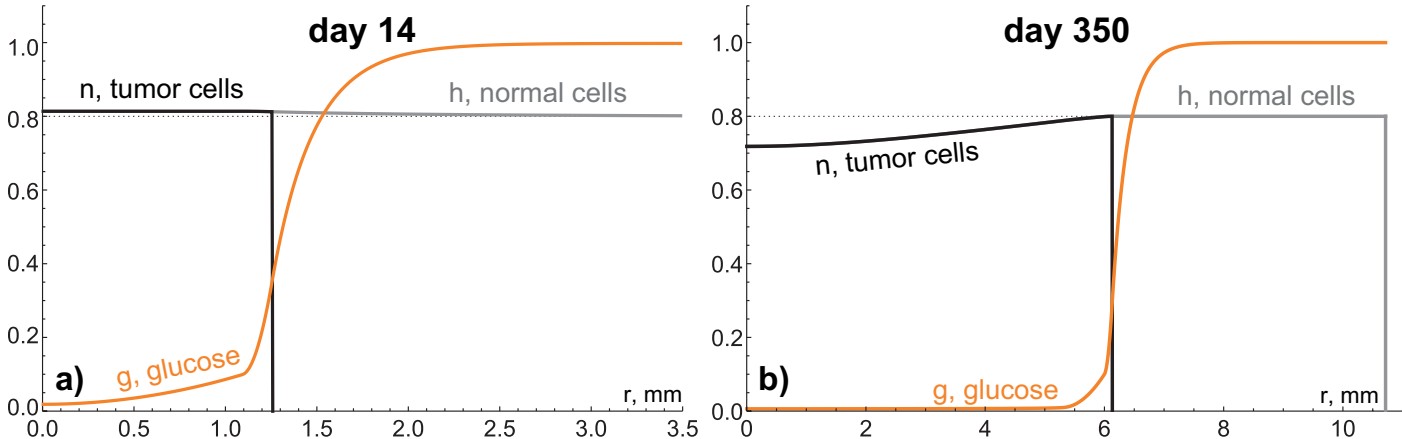

**Figure 7.** Distributions of the variables of the tumor growth model, governed by the system of Equation (6), produced by numerical simulations under tissue hydraulic conductivity $K/\mu = 0.3$ and nutrient supply level $P = 16$ on the (**a**) 14th and (**b**) 350th days of tumor growth. Dotted lines denote the value of initial fraction of cells $c_0$.

Note that Figure 7b allows deducing the qualitative pattern of cells and fluid dynamics within a stable tumor. According to Equation (6), the direction of the cell velocity throughout the tumor was opposite to the direction of tumor cell fraction gradient—since the fraction of tumor cells throughout the tumor was greater than the value $c_m \approx 0.63$, at which solid stress function would achieve its minimum. Thus, the new cells, which appeared at the outer tumor rim, moved towards the necrotic core, where they died turning into fluid. According to Equation (2), the fluid moved in opposite direction, i.e., towards the tumor boundary, where it was used again for cell proliferation.

For the largest used value of $P = 64$ the tumor growth pattern changed qualitatively, resulting in more prominent increase in maximum tumor volume, compared to the case of $K/\mu = 3$ (see Figure 6). As Figure 8a shows, around the 114th day of the tumor growth the fraction of tumor cells in the tumor center reached the value of $c_m$. Before this moment all the tumor cells located close to the tumor center had been pulled towards it. However,

upon further decrease of cell fraction in the tumor center, cells situated there began to be pulled away from it. With further tumor growth, a necrotic fluid-filled core effectively devoid of tumor cells formed within the tumor center. The transition to such a state was accompanied by complex redistribution of zones with lower and higher fractions of tumor cells within the tumor core. The details of these transient dynamics turned out to be very sensitive to the time and space steps; however, they were not investigated in detail, since after the cells in the tumor core effectively died out, the differences between simulations practically leveled out.

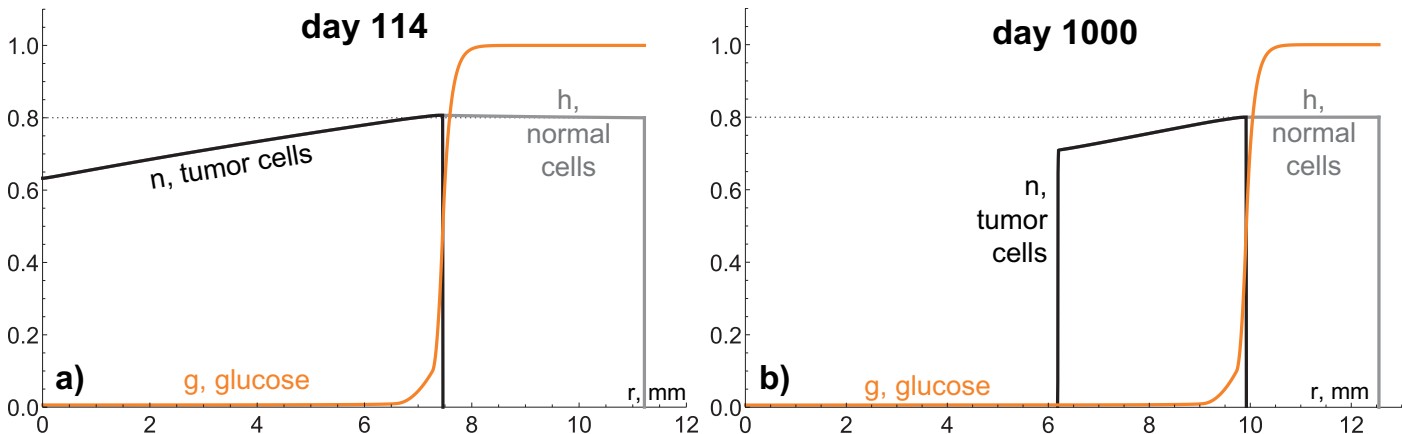

**Figure 8.** Distributions of the variables of the tumor growth model, governed by the system of Equation (6), produced by numerical simulations under tissue hydraulic conductivity $K/\mu = 0.3$ and nutrient supply level $P = 64$ on the (**a**) 114th day of tumor growth and (**b**) 1000th day of tumor growth, at which the tumor core is filled with fluid. Dotted lines denote the value of initial fraction of cells $c_0$.

The radius of the region of dying tumor cells, at which such transition should take place, can be estimated by finding the size $R_d^{cr}$, that admits the stationary solution for the following equation for dying tumor cells, obtained from Equation (6) taking into account $c_s \leq n \leq c_0$:

$$Mn = \frac{K}{\mu} \frac{k}{[c_0 - c_s]^2 r^2} \frac{\partial}{\partial r}\left(nr^2 \frac{\partial}{\partial r}([n - c_0] \cdot [n - c_s]^2)\right), \qquad (24)$$

with boundary conditions $n(R_d^{cr}) = c_0$, $n(\delta) = c_m$, $n'(\delta) = 0$, where $\delta \to 0$ and prime denotes differentiation with respect to space. For the value of $K/\mu = 3$, considered in the previous section, $R_d^{cr} \approx 21.5$ mm, which is clearly unattainable within the physiologically justified range of $P$. For the case of $K/\mu = 0.3$, considered in this section, such estimation yields $R_d^{cr} \approx 6.8$ mm, which corresponds well to Figure 8a.

Accumulation of fluid within the tumor resulted in slower deceleration of tumor growth, as Figure 6 shows for the case of $P = 64$. However, tumor growth eventually ceased in this case, since the overall rate of tumor cell death (which is approximately the volume of the region of dying cells, multiplied by the death rate $M$) grew faster than the overall rate of tumor cells proliferation. The effectively stationary distribution of the model variables for the case of $P = 64$ is illustrated in Figure 8b. Note that the abrupt declines in normal cells fractions in Figures 7 and 8 indicate the boundaries of the normal tissue, where it was displaced by tumor from its initial position of $R_0^T = 10$ mm. At that, the total volume of normal cells was conserved in every simulation, since they neither proliferated nor died.

*4.3. Low Hydraulic Conductivity: Crucial Condition for Giant Benign Tumors*

For the lower value of $K/\mu = 0.03$, the estimated value of the radius of the region of dying tumor cells, at which the formation of the fluid-filled necrotic core should begin, is $R_d^{cr} \approx 2.15$ mm. This value was achieved for each of the five used values of $P$. However, as Figure 9 shows, in the case of the lowest values of $P = 0.25$ and $P = 1$, the presence of the

fluid-filled core led to quite moderate increases of maximum tumor radii—compared to the case of $K/\mu = 0.3$, they increased by $\approx 1.5\%$ and $\approx 28.5\%$, correspondingly.

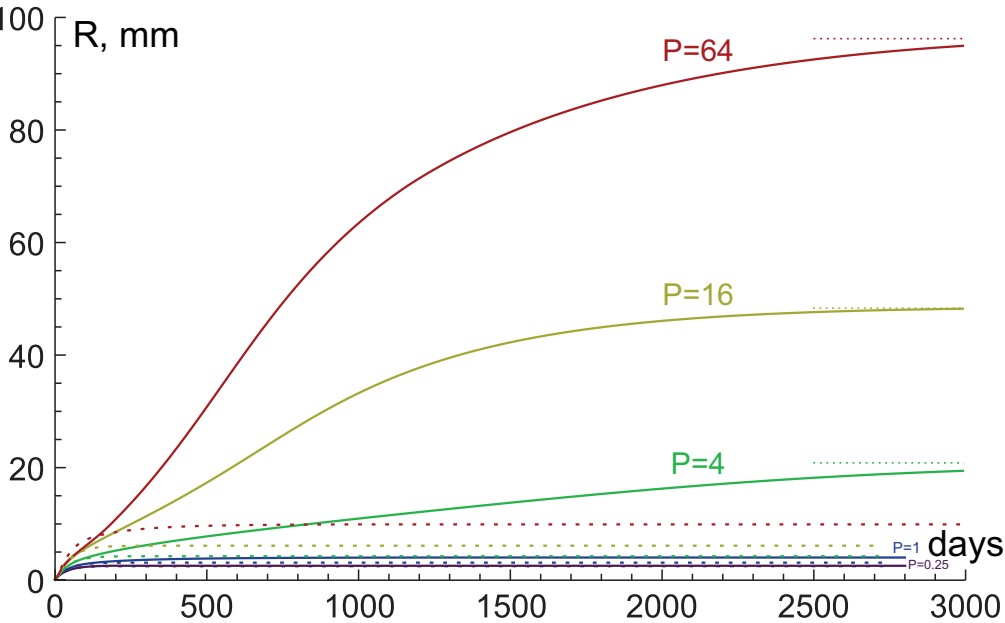

**Figure 9.** Tumor growth curves, produced by numerical simulations of the system of Equation (6) under different nutrient supply levels $P$ and tissue hydraulic conductivity $K/\mu = 0.03$ (solid lines) and $K/\mu = 0.3$ (dashed lines). Dotted lines denote the limit values of tumor radii under $K/\mu = 0.03$.

For the three greatest values of $P$, tumors grew without saturation for much longer time periods of several years, and maximum tumor volumes in these cases increased 5, 8 and 10 times, in order of increasing values of $P$. Under the greatest value of $P = 64$, the tumor grew almost up to 10 cm in radius. The growth of the three largest tumors was eventually ceased due to an effect of thinning of the normal tissue region, which did not affect the growth of the two smallest tumors. Namely, while the overall volume of proliferating cells increased with the tumor growth, the volume of the normal tissue remained constant in the simulations, as it was discussed in the previous section. Therefore, its width gradually decreased, and it eventually could not supply the tumors with a sufficient amount of glucose for continued growth. For the three largest tumors, the normal tissue width at the end of the simulations was $\approx 0.75$ mm, $\approx 0.14$ mm and $\approx 0.04$ mm (in order of increasing $P$). The maximum glucose concentration, achieved at the right boundary, was $\approx 0.58$, $\approx 0.25$ and $\approx 0.17$.

### 4.4. Very Low Hydraulic Conductivity: Stress-Induced Growth Restriction and Explosive Acceleration

Under even lower tissue hydraulic conductivity, $K/\mu = 0.003$, another effect showed up in the simulations, i.e., the restriction of tumor growth speed by stress. It happened when tumor cells fraction reached the critical value of $c_{cr} = 0.95$, that corresponded to the critical solid stress, at which cell proliferation ceased. As long as such condition held, tumor growth curve was independent of the nutrient supply level. Figure 10 illustrates this via the distributions of model variables for the same moment of time at 2000-th day of growth under $P = 4$ and $P = 64$. The two tumors had equal radii despite the obviously different total amount of tumor cells, since only a very small number of them in the tumor rims were actually proliferating.

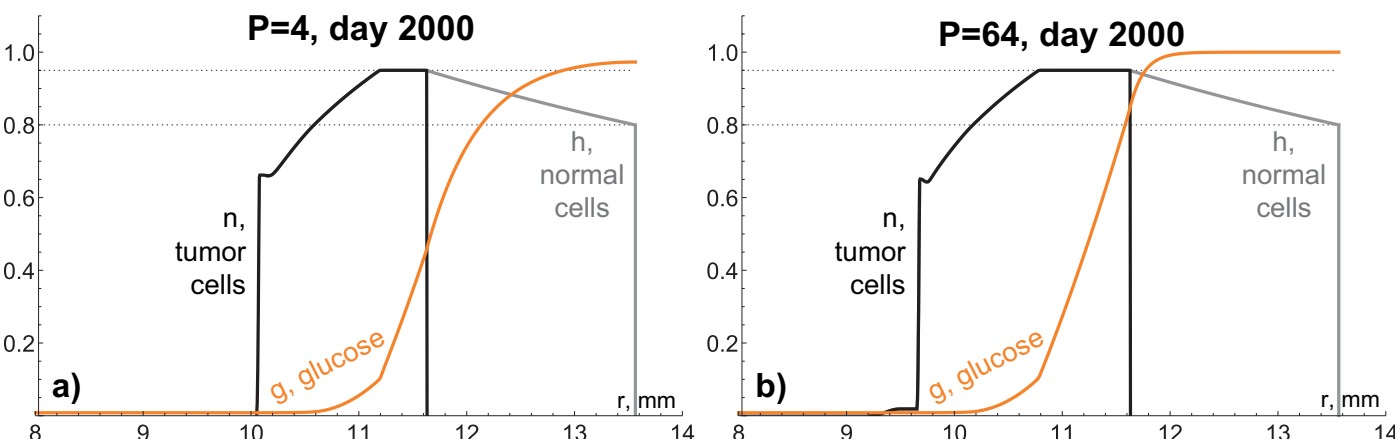

**Figure 10.** Distributions of the variables of the tumor growth model, governed by the system of Equation (6), produced by numerical simulations under tissue hydraulic conductivity $K/\mu = 0.003$ and nutrient supply levels (**a**) $P = 4$ and (**b**) $P = 64$ on the 2000th days of tumor growth. In both cases, the tumor core is filled with fluid. Dotted lines denote the values of initial fraction of cells $c_0$ and critical fraction of cells $c_{cr}$, at which cell proliferation ceases.

Tumor growth curves for the case of $K/\mu = 0.003$ are shown in Figure 11. In every case, the critical value of tumor cells fraction was already achieved on the 4th day of tumor growth, and the stress-induced growth limitation was held until different moments of time, at which the nutrient supply became insufficient to maintain the stress-induced speed limit. In the case of $P = 64$, the maximum fraction of tumor cells fell below $c_{cr}$ around the 3960th day, when the tumor radius was $\approx$63.5 mm.

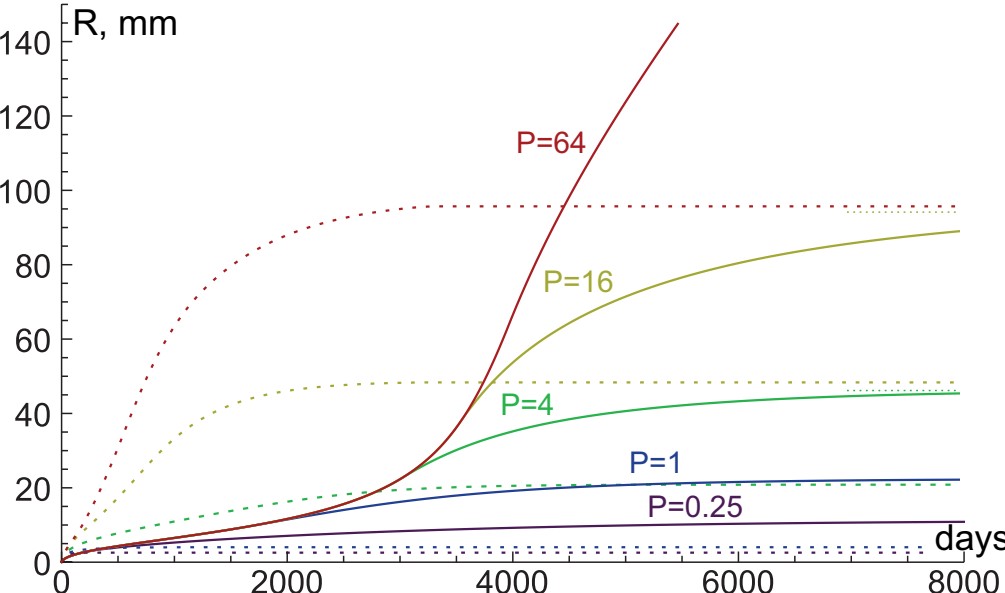

**Figure 11.** Tumor growth curves, produced by numerical simulations of the system of Equation (6) under different nutrient supply levels $P$ and tissue hydraulic conductivity $K/\mu = 0.003$ (solid lines) and $K/\mu = 0.03$ (dashed lines). Dotted lines denote the limit values of tumor radii under $K/\mu = 0.003$ and $P = 4$, $P = 16$. The simulation under $K/\mu = 0.003$ and $P = 64$ was stopped at $R = 300$ mm, before approaching stationary state.

The limit value of tumor growth speed, which can be imposed by solid stress for every value of tumor radius $R$, can be rather roughly estimated semi-analytically via finding the stationary solution for the following equation for the normal cells, obtained from Equation (6) taking into account $h \geq c_0$:

$$\frac{1}{r^2}\frac{\partial}{\partial r}\left(h(r)r^2\frac{\partial h(r)}{\partial r}\right) = 0, \tag{25}$$

whose general solution is $h(r) = C_1\sqrt{[2 - C_2 r]/r}$, with coefficients $C_1$ and $C_2$ defined by the boundary conditions $h(R) = c_{cr}$ and $h(R^T(R)) = c_0$, where $R^T(R)$ is the approximate position of the right boundary of the normal tissue, which can be found via conservation of total normal cells volume: $\int_R^{R^T} 4\pi r^2 h(r)\mathrm{d}r = \frac{4}{3}\pi(R_0^T)^3$. Then, the stress-induced limit value of tumor growth speed can be assessed as $-k\frac{K}{\mu}\frac{\partial h(r)}{\partial r}\Big|_R$.

Figure 12 demonstrates the graph of the stress-induced speed, estimated by this procedure, along with the dependence of tumor growth speed on its radius, produced by numerical simulation under $P = 64$. Starting from $R \approx 3$ mm, semi-analytical estimations resulted in the values of speed, which were by no more than a quarter lower than the numerically obtained values, both speeds reaching their minimums around $R = 6$ mm. At $\approx 47$ mm, the estimated values became greater, and around $\approx 63.5$ mm, they exceeded the numerically obtained values of tumor front speed by about a third. The further tumor growth in numerical simulation was limited by glucose diffusion and its speed fell down due to the ongoing thinning of the normal tissue. Of note, up to this point the thinning of the normal tissue on the contrary contributed to the increase of the stress-limited growth speed. This can be understood graphically via the fact that, for sufficiently large tumors, a thinner region of normal tissue with fixed values on its boundaries should have a steeper profile, which in turn should yield greater speed of its movement, as it is defined by the gradient of normal cells fraction (see Figure 10).

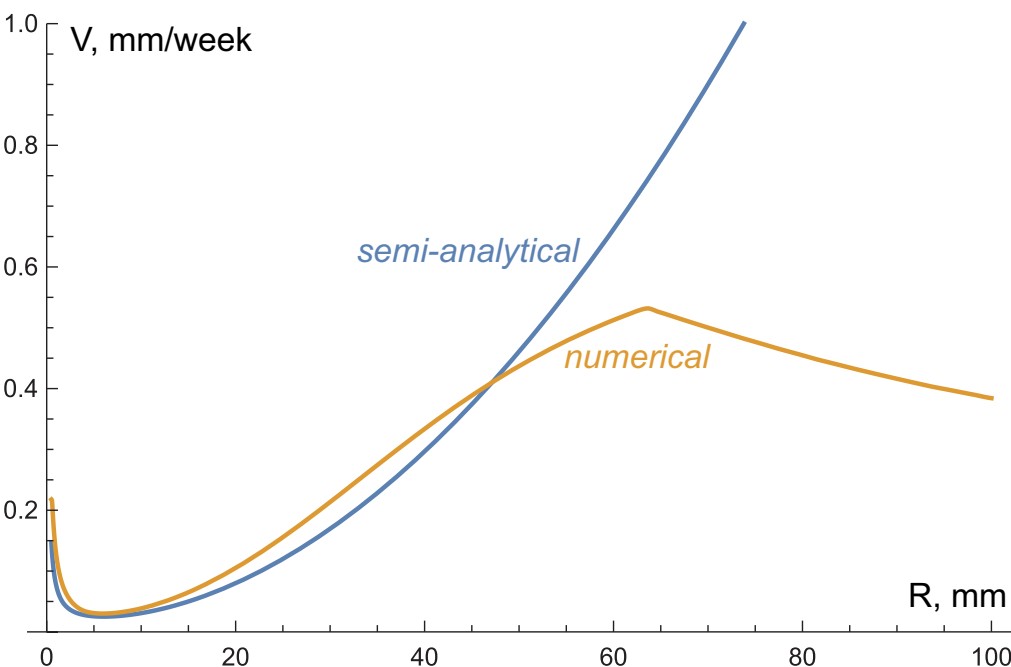

**Figure 12.** The speed of tumor growth $V$ by its radius $R$ produced by numerical simulations of the system of Equation (6) under $K/\mu = 0.003$ and $P = 64$ (orange line) and semi-analytically estimated limit values of tumor growth speed, imposed by solid stress under $K/\mu = 0.003$ (blue line).

Therefore, the volume of the normal tissue should have an ambiguous effect on the pattern of the tumor growth. This is illustrated in Figure 13, which shows tumor growth curves for $K/\mu = 0.003$ and $P = 4$ under varied volume of normal tissue. Higher normal tissue volumes initially restricted tumor growth by solid stress to a greater extent, but eventually produced longer growing tumors with greater maximum volumes.

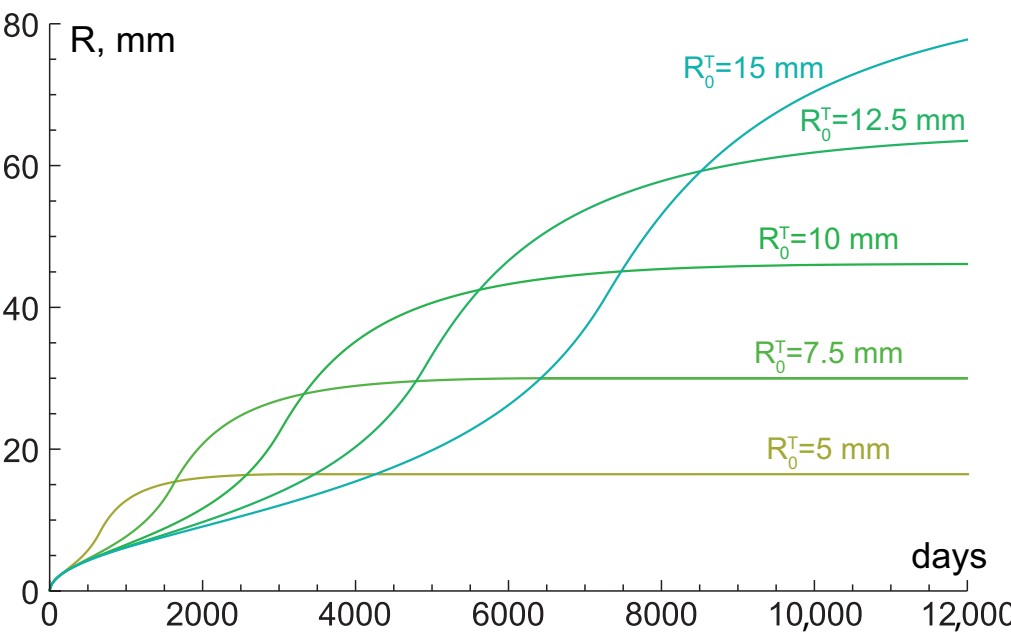

**Figure 13.** Tumor growth curves, produced by numerical simulations of the system of Equation (6) under nutrient supply level $P = 4$, tissue hydraulic conductivity $K/\mu = 0.003$ and different values of initial radius of region of normal tissue, that surrounds tumor, $R_0^T$.

## 5. Discussion

This work introduced a continuous mathematical model of spherically-symmetrical growth of a benign tumor in normal tissue. The tumor was considered to be non-invasive and avascular, the latter meaning that there were no capillaries within the tumor and it obtained nutrients only by diffusion from the surrounding normal tissue. The limitation of nutrient supply was one of the main factors that influenced tumor growth. Another factor was the biomechanical interactions of tissue elements. Tumor and normal tissue were considered as comprised of porous solid matrix and interstitial fluid, which moved through it in response to the spatial variations of fluid pressure, closely related to the spatial variations of solid stress. Solid stress was due to the repulsive and adhesive interaction of cells. Its value was considered to be linked to the local fraction of cells, which changed due to their proliferation and death.

An important approximation of the model was the limit of infinite tissue hydraulic conductivity, in which the stress gradients, arising in tissue, equalized instantaneously. It was shown that in this approximation rather accurate tumor growth curves can be obtained semi-analytically, without solving the full system numerically. Numerical simulations allowed studying the influence of variation of hydraulic conductivity $K/\mu$ and nutrient supply level $P$ on tumor growth.

### 5.1. Biological Relevance of the Results

The first problem that numerical simulations were to solve is indicating the conditions of applicability of the approximate model that assumed infinite tissue hydraulic conductivity. It turned out that in the semi-analytical solutions and simulations of the approximate model, tumor radii differed by no more than 0.5% from the ones obtained in the simulations of the full model with the basic value of $K/\mu = 10^{-8}$ cm$^2$/(mmHg$\cdot$s) within the physiologically reasonable range of nutrient supply level. Interestingly, the simulations supported the analytically obtained hypothesis, that under sufficiently high value of $K/\mu$ and sufficiently large initial radius of normal tissue region $R_0^T$ the variation of nutrient supply level should have only quite moderate effect on the tumor growth curves at their first stages. Namely, the explicit analytical formulas, obtained for infinite $K/\mu$ and $R_0^T$, suggest that at the moment of appearance of the first quiescent tumor cells, tumor radius can vary by no more than $\approx$1.7 times for varying nutrient supply level $P$ under the

basic set of parameters, and at the moment of appearance of first dying tumor cells tumor radius can vary by no more than $\approx$1.25 times. Maximum tumor radius, however, can vary by as much as $\approx$5.8 times.

The basic value of $K/\mu$ was chosen as the approximation of the lowest among the values, measured experimentally in the work [40] for four tumors of different types, grown in mice. Notably, in this work hydraulic conductivity was shown to inversely correlate with the tissue content of collagen, which is one of the main components of extracellular matrix. The lowest value of hydraulic conductivity corresponded to a certain type of sarcoma, which is a common name for tumors, that arise from connective tissue, which is rich in collagen fibers. The other three tumors, derived from epithelial and neural cells, corresponded to even greater values of $K/\mu$ by one to two orders of magnitude.

However, as it was mentioned in Section 2.3, another experimental work [46] showed that for another type of sarcoma, hydraulic conductivity can be lower than the basic value by about two orders of magnitude. Therefore, the study of the influence of decrease of $K/\mu$ on the tumor growth represented another problem that was to be solved by numerical simulations. They demonstrated that the decrease of $K/\mu$ led to a slight deceleration of tumor front propagation during the first days, but eventually produced bigger tumors. At that, two important qualitative effects were found.

The first effect was the fact that sufficiently low hydraulic conductivity and sufficiently high nutrient supply level contributed to the emergence of giant long-growing tumors with necrotic fluid accumulated in their cores. For example, the physiologically reasonable values of $K/\mu = 10^{-10}$ cm$^2$/(mmHg $\cdot$ s) (0.03 for dimensionless value) and $P \approx 0.018$ s$^{-1}$ (64 for dimensionless value) yielded a tumor with maximum radius of about 10 cm, which achieved 90% of it in $\approx$5.2 years. Interestingly, the formation of giant benign tumors in case of low hydraulic conductivity seems to be a physiologically reasonable result. Such situations were observed clinically, e.g., in cases of fibroepithelial tumors, which consist of epithelial and stromal tissues, the latter primarily being made of fibrous extracellular matrix [49,50], and in cases of fibroids, benign smooth muscle tumors, whose growth is accompanied by ample production of extracellular matrix [51,52].

The second noticeable qualitative effect that manifested itself under the lowest used values of tissue hydraulic conductivity was the inhibition of tumor growth by solid stress. In this case, tumor growth in its beginning was temporarily independent of the nutrient supply level. However, under its high values, tumor growth accelerated drastically long before reaching plateau in result of the thinning of the surrounding normal tissue. For example, under $K/\mu = 0.003$ and $P = 64$, the speed of tumor front propagation increased almost 18 times in $\approx$8.5 years after achieving its minimum around the 850-th day of tumor growth. This result suggests that significant acceleration of tumor growth may happen in case of benign tumors with low hydraulic conductivity without their progression to malignant phenotype—in particular, without alterations in intrinsic proliferation rates of individual tumor cells and without them becoming motile. Such pattern of growth is hard to observe clinically, since it requires long-term observation without intervention. However, at least several cases involving explosive acceleration of growth of benign tumors were reported for cystosarcoma phyllodes, which is a fluid-filled fibroepithelial breast tumor [53,54].

### 5.2. Prospects of the Model Development

The presented model was based on a number of assumptions, made for the sake of the possibility of analytical estimations and for proving the validity of the numerical approach. In particular, the dependencies of the proliferation rate of tumor cells on solid stress and glucose level were taken to be as simple as possible, i.e., close to step-wise functions. Upon further model development, more realistic functions will be used, based on experimental data [9,55]. Another assumption to be eliminated is the equality of properties of tumor and normal tissues, in particular, equal hydraulic conductivity values and equal dependencies of solid stress on cell fraction.

The important drawback of the current version of the model is the neglect of circumferential stress, which is the consequence of consideration of solid tissue phase as a fluid-like substance. This component of stress arises in the normal tissue and tumor periphery as proliferating parts of the tumor stretch them during their growth. Circumferential stress should squeeze the growing tumor, thus enhancing the influence of biomechanical aspects on its growth. The account for it demands a significantly more complex approach (see, e.g., [23,24]). Another aspect, consideration of which should increase the physiological correctness but also the complexity of the model, is the explicit consideration of dynamics of extracellular matrix, which is constantly produced [56] and destroyed [57] during tumor growth. It should be noted, however, that physiologically based consideration of extracellular matrix dynamics and its influence on tumor growth is a non-trivial task, that is quite rarely addressed [58].

The key task that will be focused on with the use of the developed version of the presented model is the optimization of various types of long-term tumor treatments, associated with the delivery of drugs to the tumor via intravenous injections. Consideration of biphasic tissue and the account for solid stress will not only allow reproducing adequately the dynamics of drugs and tumor during the course of therapy, but also will allow accounting for the compression of capillaries and lymphatic vessels by solid stress, which negatively affects the delivery of drugs to the tumor [13].

**Supplementary Materials:** The following are available online at https://www.mdpi.com/article/10.3390/math9182213/s1, File S1: Kuznetsov2021-program_code.cpp.

**Funding:** This work is supported by the Ministry of Science and Higher Education of the Russian Federation: agreement No. 075-03-2020-223/3 (FSSF-2020-0018).

**Conflicts of Interest:** The author declares no conflict of interest.

## Appendix A. Estimations of Glucose Distributions for the Tumor Growth Curves Approximations

In the infinite hydraulic conductivity limit, in case of $\epsilon \to \infty$ the equation for glucose distribution in different regions, obtained from the system of Equation (7), are as follows in quasistationary approximation:

$$\text{normal cells region:} \quad Pc_0[1-g] + \frac{D_g}{r^2}\frac{\partial^2(gr^2)}{\partial r^2} = 0, \ r > R,$$

$$\text{proliferating cells region:} \quad \frac{D_g}{r^2}\frac{\partial^2(gr^2)}{\partial r^2} - Q_p c_0 = 0, \ R > r > R_p,$$

$$\text{quiescent cells region:} \quad \frac{D_g}{r^2}\frac{\partial^2(gr^2)}{\partial r^2} - Q_q c_0 = 0, \ R_p > r > R_d,$$

$$\text{dying cells region:} \quad \frac{D_g}{r^2}\frac{\partial^2(gr^2)}{\partial r^2} = 0, \ R_d > r > 0.$$

At the moment $T_p$, corresponding to the transition from the first to the second growth stage, depicted in Figure 2, $R_d(T_p) = R_p(T_p) = 0$ and only proliferating cells region and normal cells region are present. At the left border $g(0) = g_p$. Here and further normal cells region is infinite and the right boundary condition for glucose is $\lim_{r\to\infty} g(r) = 1$. The glucose distribution thus is governed by the following system:

$$\text{proliferating cells region:} \quad g_I(r) = g_p + \frac{c_0 Q_p r^2}{6D_g}, \ R(T_p) > r > 0,$$

$$\text{normal cells region:} \quad g_{II}(r) = 1 + \frac{C_1}{r} exp(\frac{c_0 P}{D_g}r), \ r > R(T_p),$$

$$\text{stitching:} \quad g_I(R(T_p)) = g_{II}(R(T_p)), \ g'_I(R(T_p)) = g'_{II}(R(T_p)).$$

The stitching procedure yields the system of two equations for two variables: $R(T_p)$ and $C_1$. It has three solutions, in one of which $R(T_p)$ is real and therefore has physical meaning, its formula is given in Equation (10).

At the moment $T_d$, corresponding to the transition from the second to the third growth stages, depicted in Figure 3, $R_d(T_d) = 0$ and three regions are present. At the left border $g(0) = g_d$. Thus:

$$\text{quiescent cells region:} \quad g_I(r) = g_d + \frac{c_0 Q_q r^2}{6 D_g}, \quad R_p(T_d) > r > 0,$$

$$\text{proliferating cells region:} \quad g_{II}(r) = \frac{c_0 Q_p r^2}{6 D_g} - \frac{C_2}{r} + C_3, \quad R(T_d) > r > R_p(T_d),$$

$$\text{normal cells region:} \quad g_{III}(r) = 1 + \frac{C_4}{r} exp(\frac{c_0 P}{D_g} r), \quad r > R(T_d),$$

$$\text{stitching:} \quad g_I(R_p(T_d)) = g_{II}(R_p(T_d)) = g_p, \quad g_I'(R_p(T_d)) = g_{II}'(R_p(T_d)),$$
$$g_{II}(R(T_d)) = g_{III}(R(T_d)), \quad g_{II}'(R(T_d)) = g_{III}'(R(T_d)).$$

The formula for $R_p(T_d)$ (see Equation (16)) is obtained straightforwardly form the condition $g_I(R_p(T_d)) = g_p$. The stitching procedure yields four more equations for four variables: $R(T_d), C_2, C_3, C_4$. The physiologically meaningful solution for $R(T_d)$ is given in Equation (14).

In the limit $t \to \infty$, corresponding to the stable state, depicted in Figure 4, four regions are present. At the border between dying and quiescent cells regions $g(\hat{R}_d) = g_d$. At the left border $g'(0) = 0$. Thus:

$$\text{dying cells region:} \quad g_I(r) = g_d, \quad \hat{R}_d > r > 0,$$

$$\text{quiescent cells region:} \quad g_{II}(r) = g_d + \frac{c_0 Q_q [r - \hat{R}_d]^2 [r + 2\hat{R}_d]}{6 D_g r}, \quad \hat{R}_p > r > \hat{R}_d,$$

$$\text{proliferating cells region:} \quad g_{III}(r) = \frac{c_0 Q_p r^2}{6 D_g} - \frac{C_5}{r} + C_6, \quad \hat{R} > r > \hat{R}_p,$$

$$\text{normal cells region:} \quad g_{IV}(r) = 1 + \frac{C_7}{r} exp(\frac{c_0 P}{D_g} r), \quad r > \hat{R},$$

$$\text{stitching:} \quad g_{II}(\hat{R}_p) = g_{III}(\hat{R}_p) = g_p, \quad g_{II}'(\hat{R}_p) = g_{III}'(\hat{R}_p),$$
$$g_{III}(\hat{R}) = g_{IV}(\hat{R}), \quad g_{III}'(\hat{R}) = g_{IV}'(\hat{R}),$$

where the stitching at $r = \hat{R}_d$ is already performed. The remaining stitching procedure yields five more equations for six variables: $\hat{R}_d, \hat{R}_p, \hat{R}, C_5, C_6, C_7$. They result in the relations, expressed in Equations (20) and (21).

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
