# Peer review of "Combined Influence of Nutrient Supply Level and Tissue Mechanical Properties on Benign Tumor Growth as Revealed by Mathematical Modeling"

_mathematics, doi:10.3390/math9182213_

Round 1

Reviewer 1 Report

Report on paper "Combined influence of nutrient supply level and tissue mechanical properties on benign tumor growth as revealed by mathematical modeling" submitted by Kuznetsov, for publication in Mathematics.

The authors presented a continuous mathematical model of non-invasive avascular tumor growth in tissue in order to study the influence of combined variation of tissue hydraulic conductivity and nutrient supply level on tumor growth. The paper is interesting, globally well organized and written, it can be improved by addressing the following comments:

  1. In the introduction, the authors should give more significant details while describing the previous models of the literature.
  2. In line 64, the author should mention "to our knowledge" in the statement.
  3. In section 2, the used assumptions should be justified or clearly discussed.
  4. In subsection 5.2, it is not clear how the author intend to improve the quantitative accuracy of the model in the future work while keeping its simplicity.

Author Response

Dear reviewer, thank you for your response.

Point 1. “In the introduction, the authors should give more significant details while describing the previous models of the literature.”

Response. I have augmented the review of the existing models. Some new cites have also been added.

Point 2. “In line 64, the author should mention "to our knowledge" in the statement.”

Response. I have added it.

Point 3.  “In section 2, the used assumptions should be justified or clearly discussed.”

Response. I have expanded this section, providing more relevant information about the assumptions. Some of the assumptions are used merely for simplicity or for the sake of analytical tractability, which is clearly indicated in every case.

Point 4.  “In subsection 5.2, it is not clear how the author intend to improve the quantitative accuracy of the model in the future work while keeping its simplicity. ”

Response. There is actually no intention to increase the quantitative accuracy of the model and I as well doubt that such task can be performed without increasing the simplicity of the model. On the contrary, this model is planned to yield qualitative results – first of all, in the tasks of optimization of various types of antitumor treatments. Since the planned tasks involve numerical optimization, a lot of simulations will have to be done in order to obtain the approximate solution for each of the parameter set (see for example our previous work published in this journal “Kuznetsov, Maxim, and Andrey Kolobov. "Optimization of dose fractionation for radiotherapy of a solid tumor with account of oxygen effect and proliferative heterogeneity." Mathematics 8.8 (2020): 1204.”). Moreover, ideally the parameters will have to be varied widely in order to simulate different patients and to obtain qualitative recommendations of treatment optimization, that should be beneficial for a broad range of patients. Therefore, the relatively low computational cost of the model is crucial for achieving of such goals. 

Reviewer 2 Report

In the article entitled "Combined influence of nutrient supply level and tissue mechanical properties on benign tumor growth as revealed by mathematical modeling", the author proposed a continuous mathematical model of non-invasive avascular tumor growth in tissue. The model considers tissue as biphasic material, comprised of solid matrix and interstitial fluid and accounts for glucose as the crucial nutrient, supplied from the normal tissue, and can reproduce both diffusion-limited and stress-limited tumor growth.

I think that the mathematical development is adequate and that the article is interesting. The author performs numerous simulations and comparisons. On the other hand, I would like the author to explain if the semi-analytical model does not include the change produced at 63.5 mm in Figure 12 and how it could be included.

Finally, as a summary of the above, I believe that the article is acceptable for publication if the author explains the question mentioned above.

Author Response

Dear reviewer, thank you for your response.

Unfortunately, the radius, at which this change happens, cannot be estimated reasonably by semi-analytical approach. Certainly, some estimations can be done, but they can be only very rough, associated with a great level of uncertainty, and their numerical part is quite challenging. Let me explain this in detail.

First of all, let me remind that the semi-analytical estimations, the results of which are illustrated in Fig. 12, are obtained without consideration of distribution of glucose and tumor cells. These are estimations of only the maximum possible speed of a tumor, that is imposed by solid stress – that means that at any radius tumor may be subject to the level of glucose supply, that by itself should lead to its greater speed, but this speed cannot be reached due to another – stress – limitation. Therefore, such estimation has to deal only with the distribution of normal cells, which has an analytical solution in quasistationary approximation. That makes such estimation doable. However, the use of quasistationary approximation is by itself a rather coarse assumption, that results in ~30% discrepancy between analytical and numerical values of tumor growth speed at the point, where the abovementioned qualitative change in growth pattern takes place.

So, at this moment, the values of two speeds – the limit speed imposed by solid stress and the speed that can be maintained due to the inflow of glucose upon neglect of stress limitation – should equalize. In order to assess the latter, one may use the method of finding the glucose distribution by stitching its part from different regions as it was performed in Section 3. However, while such method is applicable in case of infinite (or at least large enough) values of tissue hydraulic conductivity, it cannot be straightforwardly applied under its low values, when the distributions of cells largely deviate from c_0. The problem is that such method requires finding the distributions of glucose and cells in quasistationary approximation (which in case of cells is already a crude assumption, as it was discussed above). In case of proliferating and dying cells such functions in general form cannot be found analytically. Moreover, the distribution of glucose in the normal cells region also cannot be found analytically under such non-uniform distribution of normal cells. This can be overcome, e.g., by consideration of constant functions of cell distribution for each region (second crude assumption) – for proliferating cells region, which width is small, a value close to $c_{cr}$ should be taken, since at the considered moment of time this value should be reached at the tumor surface. For normal and quiescent cells it is unclear, what value to choose, but it may be reasonable to start with $[c_0+c_{cr}]/2$.

While these assumptions yet seem allowable for the estimations, the next problem significantly complicates them. In the infinite hydraulic conductivity limit the rate of increase of tumor mass could be assessed as the difference between the rate of proliferation of cells and their death (see section 3.3), since the proliferating cells used up all the fluid, generated in result of cell death plus some amount of fluid, flowing into the tumor from the normal tissue. The latter part of fluid provided the increase in tumor volume. However, in the case when tumor has the fluid-filled core, only part of the fluid, produced in result of cell death, is used by proliferating cells, while the rest is left in the core. The proportions are unknown, moreover, we cannot reasonably estimate the total rate of cell death, since there is no analytical solution for the quasistationary approximation of dying cells profile. (Of note, all the quasistatinary approximations can certainly be found numerically, but in this case there is little analyticality in such approach, we just replace one straightforward numerical solution by another one, rather crude due to the quasistationary assumption. Moreover, there is a certain numerical difficulty upon using such approach, which I will highlight below).

We can nevertheless assume that all fluid, generated in result of cell death, stays within the fluid-filled region (third and very crude assumption, leading to significant overestimation of speed). Then the speed of tumor growth can be linked with the rate of total cell proliferation (analogically to how is was done in section 3.2). However, we should search for the glucose distribution accounting for the fact that the normal tissue width significantly decreases during tumor growth, and we cannot replace it by infinite region. We should therefore use the boundary condition of zero flux on the outer surface of normal tissue (the glucose concentration there falls drastically below 1 – to ~0.4 – at the moment of qualitative switch in tumor growth pattern). This leads to a numerical difficulty with finding the coefficients in the general solution of glucose profile in the normal tissue region – this procedure  is associated with large numerical errors since the general solution includes two exponential functions with positive and negative exponents.

Given all the difficulties and uncertainties, when I have realized such approach, it yielded the function of tumor speed, that can be provided by nutrient inflow, slightly decreasing over time, staying near the value ~0.82 mm/week. This is indeed an overestimation, since in numerical simulations such switch happens at the speed ~0.55 mm/week. Interestingly, the obtained function, however, would cross the semi-analytical graph of stress-induced speed limitation at ~66 day, close to where it indeed happens. But that is a mere coincidence – this graph demonstrates rapid grows in that area, therefore, any estimation, except for very large values (which are rather impossible to obtain, see below), could be considered as close to reality.

A much simpler approach can be suggested for estimation of only the maximal tumor speed, that can be in principle provided by glucose inflow for sufficiently large tumor radius, upon neglect of not only stress-induced limitation, but also of the usage of fluid generated in result of cell death for cell proliferation, and considering infinite normal tissue region (which is analogical to infinite hydraulic conductivity limit, as in Section 3, but with the rate of cell death M=0). In this approach we can merely consider the planar case and find the corresponding speed of tumor propagation. If we take cell fraction to be constant and equal to c_0 (which is reasonable, since we use the variation of infinite hydraulic conductivity limit), then we obtain the maximum speed of ~0.92. It can be therefore suggested that the switch in growth pattern should happen before this value of speed is achieved. However, this is a very rough approximation with questionable usefulness, moreover, it is clearly not relevant at all for lower values of nutrient supply level.

Overall, due to the complexity of the estimation procedure, a lot of associated uncertainties and due to the fact that it requires a lot of space to be described, yielding only very crude estimations, I would not like to include it in the article.

Reviewer 3 Report

The paper by Kuznetsov deals with the interplay of tissue mechanical properties and nutrient supply level, in the framework of avascular benign solid tumor research. By means of theoretical formulation and numerical simulations, the roles of the different parameters into play are investigated. After recalling the physiological setting, and introducing the spherically-symmetric biphasic (porous matrix and interstitial fluid) model, the author identifies the limit of infinite hydraulic conductivity as amenable to analytical resolution, and describes the three consequent growth stages. Focusing then on a physical situation away from the previous ideal one, he shows the importance of glucose concentration and stress balance, and assesses the biological implications and future developments. His results are carefully presented and plotted, and the relevance of these findings with respect to the existing bibliography from scientific literature is thoroughly discussed.
In my opinion, the article deserves publication on Mathematics. Hereafter I only mention a few minor issues and typos, to be clarified or amended by the author.

1) Line 10: "at" -> "in"

2) Line 52: "this" -> "these"

3) Lines 118, 465 and 539: "which" -> "whose"

4) Line 198: "suggests" -> "suggest"

5) Lines 245-246: an aside begins with a comma and ends with a dash, please uniformize the notation.

6) Line 265: "and" -> "a"

7) Line 455: "numbers" -> "number"

8) Line 480: "more steep" -> "steeper"

9) The author uses both the present tense and the simple past in the manucript, please pay attention to temporal consistence.

Author Response

Dear reviewer, thank you for your remarks. I have corrected the corresponding issues and typos and I have corrected the misuse of the present tense, using past tense for description of the actions that have been performed during the study, e.g., numerical simulations, and leaving present tense only for objective facts, like behavior of certain functions, or contents of figures (when I submitted my first paper to this journal, I used present tense everywhere, and the editors corrected it throughout the paper – so now I have tried to use past tense for description of the results, but since this was unusual for me, I accidentally made some mistakes).

Reviewer 4 Report

In this paper, a continuous mathematical model of non-invasive avascular tumor growth is emphasized. The model considers tissue as biphasic material, comprised of solid matrix and interstitial fluid. The convective motion of tissue elements are possible due to the gradients of stress, which changes in result of tumor cells proliferation and death. The model accounts for glucose as the crucial nutrient, supplied from the normal tissue, and can reproduce both diffusion-limited and stress-limited tumor growth. Approximate tumor growth curves are obtained semi-analytically in the limit of infinite tissue hydraulic conductivity, that implies instantaneous equalization of arising stress gradients. These growth curves correspond well to the numerical solutions and are represented by classical sigmoidal curves with a short initial exponential phase, subsequent almost linear growth phase and a phase with growth deceleration, at which tumor tends to reach its maximum volume. The influence of two model parameters on tumor growth curves is investigated: tissue hydraulic conductivity, that links the values of stress gradient and convective velocity of tissue phases, and tumor nutrient supply level, which corresponds to different permeability and surface area density of capillaries in the normal tissue that surrounds the tumor. In particular, it is demonstrated, that sufficiently low tissue hydraulic conductivity (intrinsic, e.g., to tumors arising from connective tissue) and sufficiently high nutrient supply can lead to formation of giant benign tumors, reaching several centimeters in diameter, which can be clinically observed.

The paper is structured as follows:

1. In the Introduction, the author emphasizes the general context for the present study. He states that the models used in oncology, based mainly on approaches from solid mechanics, can yield more realistic reproduction of solid stress distribution within the tissue and can even provide quantitative predictions that are consistent with several experimental results. However, their solution is associated with much greater computational costs and they are not amenable to analytical investigation, like simpler approaches. It should be noted, that in general, modeling tumor growth with account of biomechanical properties is not a very popular area. One of the unexplored topics is the combined influence of both crucial growth-limiting factors – nutrient availability and mechanical stress – on tumor growth;

2. The second section emphasizes the Model described in this paper, which considers growth of a non-invasive avascular tumor in biphasic tissue and allows a semi-analytical investigation of the tumor growth in the limit of infinite tissue hydraulic conductivity. Several issues are approached, such as: Full system, Infinite hydraulic conductivity limit, Parameters, Numerical solving;

3. The third section is dedicated to the Estimation of growth curves in the limit of infinite tissue hydraulic conductivity, highlighting following steps: First growth stage: exponential phase, Second growth stage: two-layered tumor, Third growth stage: tending to plateau and Comparison with numerical simulations;

4. The forth section is concerned with the Study of influence of tissue hydraulic conductivity and nutrient supply level on tumor growth. Following cases are studied: High hydraulic conductivity: when the infinite hydraulic conductivity limit is applicable, Low hydraulic conductivity: crucial condition for giant benign tumors and Very low hydraulic conductivity: stress-induced growth restriction and explosive acceleration;

5. In the last section, dedicated to Discussions, the author presents a short version of the results obtained in this paper. The Biological relevance of the results is emphasized, as well as Prospects of the model development. The key task that will be focused on the use of the developed version of the given model is the optimization of various types of long-term tumor treatments, associated with the delivery of drugs to the tumor via intravenous injections. Consideration of biphasic tissue and the account for solid stress will not only allow to adequately reproduce the dynamics of drugs and tumor during the course of therapy, but also will allow to account for the compression of capillaries and lymphatic vessels by solid stress, which affects negatively the delivery of drugs to the tumor.

6. Other results obtained in the domain are indicated in the generous section dedicated to the References, which put an end to the paper.

The paper is of scientific interest, being well organised, written and documented. It emphasizes the importance of mathematical modeling in approaching various domains of the real world, such as, in this particular case, oncology. The terminology and the methods presented in this paper are consistent with the standards of the journal.

In my opinion, the paper deserves to be published.

Author Response

Dear reviewer, thank you for your response. As there are no remarks, it did not lead to alterations in the manuscript.

Reviewer 5 Report

{\bf Summary}

The paper studies a problem that has not garnered quite some interest yet, but has some potential. Namely, it considers a model of benign avascular tumor growth depending of two parameters -- a nutrient (glucose) level and a level of a mechanical stress. A thorough analysis of the model is provided, the model itself assumes strong additional restrictions: (i) normal cells are considered to be inert (!); (i) nutrient supply is distributed by normal cells only; (iii) circumferential stress is marginal. 

{\bf Motivation and Related Work}

An appropriate motivation and overview of known results is given.

{\bf Important questions to address}

In many cases, formulas appear without any derivations which makes it impossible to check their correctness. It may be a good idea to show where these formulas come from in the appendix. Some transitions between formulas are not quite clear as well. E.g., why the sum of the equations from (1) gives us (2).

What if we initially have no tumor cells ($n = n_0 = 0$) or no normal cells ($h = h_0 = 0$)? Do you consider such cases at all? If not, it would be nice to have the reasons described in a discussion section. 

The assumption that ``normal cells are considered to be inert'' looks a bit too strong (and implausible). No wonder that under such assumption the fraction of cancer cells remains moderate. 

It is said that only benign tumors are considered, however, it is not clear what the difference it would make to suppose that the tumor is malignant (under the model assumptions).

In the section 5.2 (lines 559--560), the author claims that ``changes should
not affect the qualitative results, obtained in this study''. It looks very questionable to me, I am actually almost certain that the results will change significantly if you remove any of the assumptions (i), (ii), or (iii).

{\bf Presentation}

Minor language problems, nothing that hinders understanding. E.g., throughout the paper, the author uses the expression ``allow to do something'', which is incorrect: you may say ``allow me/us to do'' or ``allow doing'', but not ``allow to do''.

I would encourage the author in making figure captions more comprehensive describing in the caption what is on the picture. For now one can hardly understand what is going on in the figures without thorough reading.  

I liked the style of coding -- with appropriate comments and structure. A style of the article itself leaves a good impression; I have spotted only a few occasional typos, like, e.g., in formula (8) $\leq$ should be used instead of $<=$. 

{\bf Resume}

The model, even though not that simple, works under the strong assumptions that make it a bit unrealistic. We may consider it a first step in this direction, however, I have some slight reservations about its potential. It may well happen that having removed some of the  assumptions (i), (ii), and (iii), we will end up with a very challenging model which will be hard to analyze in both ways: analytically and numerically. Despite this, I am inclined to recommend the manuscript for publication providing my aforementioned comments are properly addressed. 

Author Response

Dear reviewer, thank you for your response.

Point 1. In many cases, formulas appear without any derivations which makes it impossible to check their correctness. It may be a good idea to show where these formulas come from in the appendix.

Response. This remark must be about the formulas for tumor radius and coordinates of border between different region, obtained in the result of the stitching procedure of the function of glucose distribution at certain moments of time. Previously I have only described the essence of the method and provided a reference to my previous work, where it was used. Now I have added the appendix, where I have described this procedure in more detail in relevance to the considered problem.

Point 2. Some transitions between formulas are not quite clear as well. E.g., why the sum of the equations from (1) gives us (2).

Response. I have clarified this transition in the text.

Point 3. What if we initially have no tumor cells ($n = n_0 = 0$) or no normal cells ($h = h_0 = 0$)? Do you consider such cases at all? If not, it would be nice to have the reasons described in a discussion section.

Response. If there are no normal cells in the beginning, then the tumor will not grow, since it needs glucose for growth, and it is supplied from the normal tissue. If there are no tumor cells in the beginning and the other initial conditions are as they were, then the system will remain in this stable state. Sufficiently small deviations will relax to it, however, large deviations of normal cells distribution, that involve local concentrations below the cell fraction, at which solid stress is minimal, may settle in another stable states. In such cases, there indeed may be some interesting and non-trivial dynamics and probably even chaotic behavior (since numerical simulations of the formation of fluid-filled core in section 4.2 suggest that). In my point of view, investigation of such dynamics represents an intriguing problem, which is however pretty far from the questions posed in this paper. However, it is clear that it requires more intricate numerical methods – as I wrote in the manuscript, “this transient dynamics turned out to be very sensitive to the time and space steps” and it may turn out, that this problem cannot be adequately reproduced by continuous methods at all. Moreover, description of such problem requires a thorough research on the topic of collective cell dynamics – there are clearly a lot of works, devoted to analogical questions, that cover different aspects (see, e.g., “Albert P. J., Schwarz U. S. Dynamics of cell ensembles on adhesive micropatterns: bridging the gap between single cell spreading and collective cell migration //PLoS computational biology. – 2016. – Т. 12. – №. 4. – С. e1004863.” or “Moure A., Gomez H. Phase-field modeling of individual and collective cell migration //Archives of Computational Methods in Engineering. – 2021. – Т. 28. – №. 2. – С. 311-344.”). Therefore, it seems to me that honest discussion of such separate problem will require a lot of explanation and will only lead away from the main questions posed in the paper. 

Point 4. The assumption that ``normal cells are considered to be inert'' looks a bit too strong (and implausible). No wonder that under such assumption the fraction of cancer cells remains moderate.

Response. There should be some misunderstanding here, which should be the result of the fact that the word “inert” actually does not fit well where I used it. Thank you for indicating that and let me explain this moment. The normal cells are considered to not proliferate and not consume glucose, contrary to the tumor cells (or at least the rates of these processes are negligible) . Also their death due to the lack of glucose is not considered in the equations, since such low levels of glucose cannot be achieved in the normal tissue during the performed simulations. From that it follows, that the total number of normal cells is conserved (as well as their volume, due to incompressibility). Further, as well as the tumor cells, normal cells lack intrinsic motility. However, their interaction as well generates solid stress, and they move passively due to the arising convective flows (so they are actually not quite inert). I have now removed the word “inert” from the manuscript to avoid such misunderstanding.

However, the fraction of cancer cells does not always remain moderate. The total volume of the normal cells is ~3.3 cm^3, as initially they occupy 80% of volume of a sphere of 1 cm in radius. As follows from Figure 5, corresponding to infinite hydraulic conductivity limit – in this case tumors do not have fluid-filled core and their fraction is homogeneous, equal to 0.8 – under sufficiently high nutrient supply level the total volume of tumor cells even in this case can achieve that of normal cells and even exceed it. And when hydraulic conductivity is finite, the total volume of tumor cells can significantly exceed that of normal cells in case of giant benign tumors – for example, in the case with P=64, illustrated in Figure 9, the total volume of tumor cells is ~123 cm^3 in the end of the simulation – so tumor cells are actually ~97% of all cells in the considered region.

Point 5. It is said that only benign tumors are considered, however, it is not clear what the difference it would make to suppose that the tumor is malignant (under the model assumptions).

Response. The difference between benign and malignant tumors has already been discussed in the Introduction. The most crucial difference between them is invasion of nearby tissues, which, as it has been written, “is a direct indicator of malignant cancer”. The simplest and most popular way to account for it (at least, when mechanical aspects are ignored) is to introduce the intrinsic motility of cancer cells and to include their movement via diffusion-like term. See, e.g., my previous work “Kuznetsov M. Mathematical Modeling Shows That the Response of a Solid Tumor to Antiangiogenic Therapy Depends on the Type of Growth //Mathematics. – 2020. – Т. 8. – №. 5. – С. 760.”. There exists ample literature on modeling high-invasive tumors (most often glioblastomas) via reaction-diffusion approach with neglect of convective terms. There certainly exist other differences between benign and malignant tumors, but they mostly correspond to the degree of manifestation of another hallmarks of cancer – e.g., malignant tumors almost always induce active angiogenesis, but some benign tumors also stimulate it. This question is highlighted, e.g., in the paper that I have cited by #7 (“Lazebnik, Y. What are the hallmarks of cancer? Nature Reviews Cancer 2010, 10, 232–233.”)

Point 6. In the section 5.2 (lines 559--560), the author claims that ``changes should not affect the qualitative results, obtained in this study''. It looks very questionable to me, I am actually almost certain that the results will change significantly if you remove any of the assumptions (i), (ii), or (iii).

Response. There is also a misunderstanding – in that phrase I meant only the alleviation of the assumptions that have been just mentioned above, i.e., the change of the dependencies of the proliferation rate of tumor cells on solid stress and glucose level. Removal of assumptions (ii) or (iii) should indeed affect the results (the first assumption was discussed in point 4, where I indicated another misunderstanding). To avoid the ambiguity, I have now eliminated that phrase.

Point 7. Minor language problems, nothing that hinders understanding. E.g., throughout the paper, the author uses the expression ``allow to do something'', which is incorrect: you may say ``allow me/us to do'' or ``allow doing'', but not ``allow to do''.

Response. I have corrected it.

Point 8. I would encourage the author in making figure captions more comprehensive describing in the caption what is on the picture. For now one can hardly understand what is going on in the figures without thorough reading. 

Response. I tried to improve the figure captions, making them more comprehensible without thorough reading of the text. 

Point 9. I liked the style of coding -- with appropriate comments and structure. A style of the article itself leaves a good impression; I have spotted only a few occasional typos, like, e.g., in formula (8) $\leq$ should be used instead of $<=$.

Response. Thank you. I have corrected it.

Point 10. It may well happen that having removed some of the  assumptions (i), (ii), and (iii), we will end up with a very challenging model which will be hard to analyze in both ways: analytically and numerically.

Response. That is a very interesting remark. In order to answer this question with full certainty, the model with such removed assumptions must be created, which has not yet been done and is subject of the future work. However, at the moment it seems to me that it is possible to create such a model which numerical cost will be quite tolerable and suitable for the solution of tasks of numerical optimization of antitumor protocols (in which a lot of simulations have to be done in order to obtain the approximate solution for each of the parameter set). More specifically, for the account for more realistic supply of nutrients from capillaries another variable has to be introduced, that will correspond to the microvasculature density – and its properties, that are dependent on tumor’s angiogenic activity, can be accounted for via another variable, as it has been done in our previous works (see, e.g., “Kuznetsov M. B., Kolobov A. V. Transient alleviation of tumor hypoxia during first days of antiangiogenic therapy as a result of therapy-induced alterations in nutrient supply and tumor metabolism–Analysis by mathematical modeling //Journal of theoretical biology. – 2018. – Т. 451. – С. 86-100.”). Regarding the “inertness” of normal cells, there must have been a misunderstanding, which I have already discussed. However, the consumption of glucose by normal cells can be included in the model, increasing its complexity only slightly, while the assumptions of absence of intrinsic motility of normal cells and neglect of their death should still be valid (as it was discussed above, cell death happens only in the tumor core, where there are no normal cells). Consideration of circumferential stress without introducing complex and computationally expensive approaches, adapted from the area of solid mechanics, is a trickier question that should be investigated separately. Of note, there is little doubt that analytical tractability of the model will be significantly impaired by the removal of the indicated assumptions, but the goal of analytical tractability for such more complicated model will not be pursued – the main goal is the use of it for numerical optimization of antitumor protocols.

Round 2

Reviewer 1 Report

The author has addressed my comments sufficiently to recommend publication of the paper in its current form.

Reviewer 5 Report

I think that the author has done a good job revising the manuscript and adequately addressing all my comments. I recommend this paper for publication now.